# CULTURE IN ACTION: EVALUATING TEXT-TO-IMAGE MODELS THROUGH SOCIAL ACTIVITIES

**Sina Malakouti**
University of Pittsburgh
sem238@pitt.edu

**Boqing Gong**
Boston University
bgong@bu.edu

**Adriana Kovashka**
University of Pittsburgh
kovashka@cs.pitt.edu

https://sinamalakouti.github.io/AHEaD/

## ABSTRACT

Cultural nuances are best expressed through social interactions, yet current text-to-image (T2I) benchmarks focus largely on object-centric artifacts (e.g., food, landmarks, and attire). In this work, we study the *cultural faithfulness* of T2I models (i.e., adherence to the target culture) through social activities. To this end, we introduce CULTIVate, a new benchmark of 576 activities across 9 categories (e.g., dancing, greeting, dining) with over 19,000 images from 16 countries. We further propose AHEaD, an explainable framework that measures cultural understanding along four dimensions: cultural **A**lignment, **H**allucination, **E**xaggeration, and **D**iversity. Unlike prior work relying on costly human evaluation or image-text alignment (ITA), AHEaD uses culturally-grounded descriptors to provide quantitative, interpretable feedback that enables iterative image refinement. Our analysis shows ITA metrics correlate poorly with human judgments and that alignment alone is insufficient to capture *faithfulness*. In contrast, FAITH (combining alignment, hallucination, and exaggeration) achieves 27% higher correlation than baselines. Finally, we observe systematic disparities, with generated images being consistently more faithful for Global North than Global South cultures.

## 1 INTRODUCTION

The 2007 film Ratatouille earned 41 film awards including Best Feature at the 2008 Oscars (Wikipedia). Part of its appeal lies in the very realistic portrayal of the city of Paris, and of French culture and cuisine (SeattleTimes). To achieve this, creators visited places in Paris to soak in the culture and environment, including its highly distinctive visual aspects. Many other well-regarded films (animated ones like Luca and Coco, and live action ones like Amelie, Crouching Tiger Hidden Dragon, and Reservation Dogs) also devoted significant effort to ensuring they capture the true atmosphere and visuals of the places they portray. Such culturally accurate visual portrayals are important for many types of creative and marketing content beyond film, e.g., advertising.

Recent advances in text-to-image (T2I) generative models offer the promise of automating creation of such content. However, T2I models are trained on web data exhibiting strong WEIRD biases (Western, Educated, Industrialized, Rich, and Democratic) (Henrich et al., 2010), leading to incorrect or overly stereotypical cultural representations. This problem is particularly severe for social activities, where cultural meaning emerges from context, interactions, and relations between objects and people (Geertz, 2017; Hall, 1973). Despite this, cross-cultural studies of T2I models remain understudied, with existing benchmarks focusing on a few specific object-centric artifacts such as landmarks, clothing, and food (Chiu et al., 2025a; Basu et al., 2025; Rege et al., 2025).

In this work, we examine how well T2I models portray different cultures through *activities*, whose visual representations vary significantly across cultures. Unlike static artifacts, activities are **contextual and compositional**, encompassing objects, interactions, and spatial arrangements that better capture cultural expression. For example, "eating at home in Iran" may involve sitting at a table or gathering on the floor around a traditional *sofreh*– the same activity can have multiple valid cultural variants. To this end, we introduce **CULTural acTIViTy (CULTIVate)**, a culturally-grounded benchmark spanning 16 countries and 576 activities across 9 categories (e.g., dining, greeting, game, dance, celebration). We evaluate 6 state-of-the-art T2I models, generating 19,000+ images and collecting 3,000 real reference images. As shown in Fig. 1 (b), T2I models may generate wrong activi-

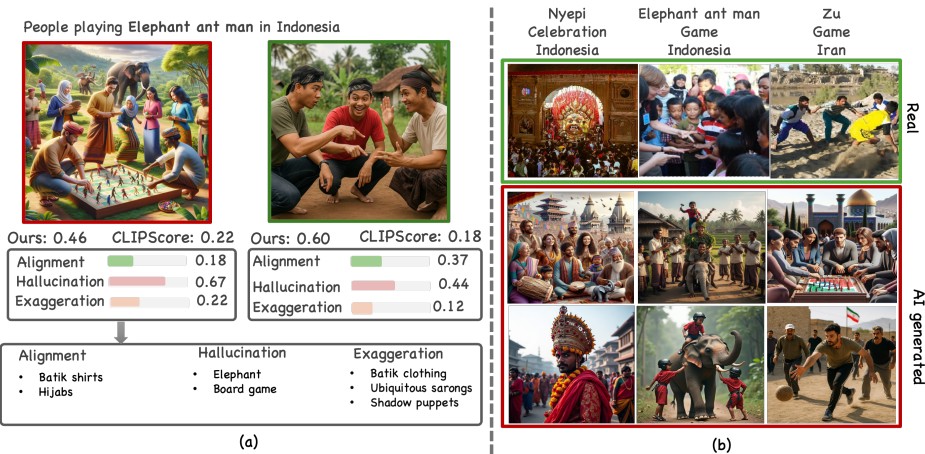

Figure 1: (a) Our framework captures alignment, hallucination, and exaggeration via interpretable descriptors. CLIPScore incorrectly scores the culturally wrong image (red border) higher due to exaggerated and hallucinated elements (e.g., elephants), while our metrics more accurately evaluate *cultural faithfulness* by capturing these 3 complementary metrics. Bottom: descriptor-level feedback identifies specific issues. (b) T2I systems generate incorrect and exaggerated images.

ties, include *hallucinated* elements, or produce *exaggerated* scenes. These complex failure patterns raise critical questions: *Are current metrics effective for evaluating cultural faithfulness? What makes a metric effective for this task?*

Our work explores *cultural faithfulness (faithfulness)*, whether images accurately represent the target culture. Prior works rely on costly human evaluation (Kannen et al., 2024; Bayramli et al., 2025), while (Rege et al., 2025; Khanuja et al., 2024) used VLM-based image-text-alignment (ITA) metrics (e.g. CLIPScore (Hessel et al., 2021)) as a proxy for human judgment of *faithfulness*. However, VLMs inherit similar cultural biases (Rege et al., 2025), exhibit poor compositional and implicit prompt understanding (e.g., bag-of-words behavior) (Yuksekgonul et al., 2023), making ITA unreliable for cultural evaluation. For example, Fig. 1(a) shows ITA metrics reward behaviors such as exaggeration and hallucination such as generating elephants for "elephant ant man game" (a rock-paper-scissors game in Indonesia). In fact, our analysis reveals that **in contrast to human judgment, ITA metrics correlate positively with exaggeration**, where more stereotypical elements increase alignment but hurt faithfulness.

To cope with these challenges and answer above questions, we introduce AHEaD (**A**lignment, **H**allucination, **E**xaggeration, and **D**iversity), a diagnostic framework using **external visual descriptors**. We decompose activities into *interpretable visual descriptors* that capture cultural elements across multiple dimensions. First, we create reference descriptors using a proposer-refiner approach where multiple proposers utilize an LLM to generate diverse candidates, and the refiner filters duplicates and errors. Second, we extract predicted descriptors from generated images using MLLMs. Unlike prior work that uses VLMs to directly score faithfulness, we use MLLMs only for generic scene understanding, avoiding reliance on their cultural biases. Finally, we compare reference against predicted descriptors to compute AHEaD metrics. AHEaD provides **interpretable insights** where *Alignment* measures cultural coverage, *Hallucination* quantifies incorrect elements, *Exaggeration* measures over-representation, and *Diversity* captures semantic variation in cultural elements. Our framework enables identifying which cultural aspects are missing, over-represented, or faithfully depicted.

In more detail, our framework provides three key capabilities. First, we compute AHEaD metrics automatically without human annotations, enabling scalable evaluation across countries and T2I models. The metrics compare different aspects of quality in the images generated by different T2I models, and can be used to quantitatively judge which T2I model to deploy when aiming to depict a particular country. Second, AHEaD outputs interpretable diagnostics including top-k and bottom-k descriptors for missing, hallucinated, and exaggerated elements, supporting targeted model improvements through descriptor-guided editing. Third, we analyze correlations between metrics to reveal trade-offs, such as whether increasing alignment affects hallucination or exaggeration.

We conduct comprehensive experiments on CULTIVate revealing systematic limitations in current cultural evaluation. We show that existing ITA-based metrics correlate poorly with human judgment. Importantly, analysis suggests AHEaD metrics are complementary, where combining Alignment, Hallucination, and Exaggeration (FAITH) achieves highest correlation than using Alignment alone. FAITH shows 27% higher correlation with human judgment of *faithfulness* than MLLM-as-judge baselines and significantly outperform ITA metrics. Finally, we find consistent bias across all T2I models, with 4-8% higher Alignment for Global North (GN) than Global South (GS) countries.

To summarize, our contributions are:

1. We introduce CULTIVate, a benchmark for evaluating cultural faithfulness of T2I models through social activities.
2. We propose AHEaD, a framework for diagnosing *cultural faithfulness* across multiple dimensions (alignment, hallucination, exaggeration, and semantical diversity) using interpretable visual descriptors that could be used for descriptor-guided image refinement.
3. Analysis showing three key findings: ITA metrics are ineffective, alignment alone is insufficient, and combining alignment, hallucination, and exaggeration is necessary, achieving best correlation with human judgments.
4. We reveal consistent bias in T2I models towards GN countries.
5. Proposer-refiner enables robust, scalable reference descriptors without human annotations.

## 2 RELATED WORKS

**Image-Text Alignment Metrics.** General-purpose metrics rely on low-level features (e.g. FID (Heusel et al., 2017), LPIPS (Zhang et al., 2018)) or global image-text alignment (e.g. CLIPScore (Hessel et al., 2021), VQAScore (Lin et al., 2024)). Some metrics require expensive human judgments (e.g. ImageReward (Xu et al., 2023), PickScore (Kirstain et al., 2023)). We show these correlate poorly with human judgment.

**Cultural Benchmarks.** Cultural understanding has been extensively studied for image understanding tasks (Kalluri et al., 2023; Ramaswamy et al., 2023; Nayak et al., 2024; Astruc et al., 2024; Vayani et al., 2025; Liu et al., 2025; Yin et al., 2023). For T2I generation, existing benchmarks are primarily object-centric (Kannen et al., 2024; Basu et al., 2023; Zhang et al., 2024; Rege et al., 2025; Liu et al., 2024; Jha et al., 2024). For instance, (Kannen et al., 2024) covers 8 countries across 3 artifact categories, (Jha et al., 2024) includes 10 countries on food and architecture, and (Basu et al., 2023) covers 27 countries using parsed noun phrases. CULTIVate differs by evaluating social activities, which are compositional and contextual, creating distinct evaluation challenges beyond object recognition (e.g., correct interaction, spatial arrangements). Concurrent work (Nayak et al., 2025) studies cultural expectations through human evaluation. We complement this by focusing on activities and proposing the first automated metrics for cultural faithfulness specialized for social activities.

**Cultural Representativeness Metrics.** Cultural representativeness is typically measured through *diversity* and *faithfulness*. Diversity-based metrics (Rege et al., 2025; Kannen et al., 2024; Basu et al., 2025; Zhang et al., 2024) quantify variation across generated outputs using approaches like continent-level diversity scores (Kannen et al., 2024; Friedman & Dieng, 2023) or perceptual similarity between images (Rege et al., 2025). However, diversity alone is insufficient for measuring cultural representativeness (Rege et al., 2025), as vision encoders exhibit geographical bias and capture low-level variations (color, texture) rather than cultural content. Importantly, **diversity and *faithfulness* are distinct**: our work focuses on faithfulness.

**Cultural Faithfulness**. Existing works (Nayak et al., 2025; Kannen et al., 2024; Jha et al., 2024; Liu et al., 2024) rely on accurate but costly and unscalable human evaluation. Recently, some studies (Khanuja et al., 2024; Basu et al., 2023; Rege et al., 2025) adopted VLMs-based image-text alignment (e.g. CLIPScore (Hessel et al., 2021)) as a proxy for human judgment on cultural faithfulness. Specifically, (Khanuja et al., 2024) measures alignment with simple country prompts, while (Rege et al., 2025) measures alignment between hierarchical prompts. However, we show that these metrics do not correlate well with human judgment (Tab. 2). These approaches rely on VLM embeddings to directly measure faithfulness. However, VLMs inherit similar cultural biases (Rege et al., 2025), struggle with compositional and implicit prompt understanding (Yuksekgonul et al., 2023;

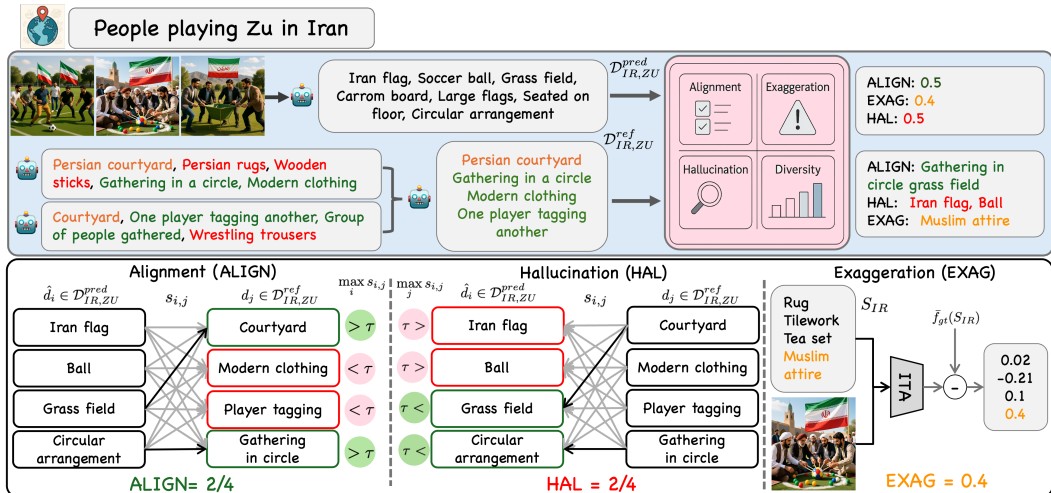

Figure 2: **(Top) Overview.** We extracted image descriptors $\hat{d}_i \in \mathcal{D}^{pred}$ with InternVL3, while reference descriptors $d_j \in \mathcal{D}^{ref}$ are obtained via a proposer–refiner pipeline in data annotation stage without using images. Proposers generate diverse candidates, and the Refiner removes duplicates and filters incorrect ones. AHEaD measures cultural competence through alignment, hallucination, exaggeration, and diversity, providing not only quantitative scores but also interpretable feedback (i.e., what is aligned, missing, or exaggerated). **(Bottom) Cultural Faithfulness metrics.** Alignment measures whether expected descriptors are present (similarity above threshold $\tau$), hallucination flags elements unsupported by references (e.g., circular arrangement), and exaggeration detects exaggerated cues overemphasized with respect to real-images (e.g., Muslim attire)

Li et al., 2025; Malakouti et al., 2025; Aghazadeh & Kovashka, 2025), making ITA unreliable for cultural evaluation. We introduce AHEaD, a suite of automatic metrics that leverages *external visual descriptors* to measure cultural alignment while penalizing hallucinations and over-exaggeration.

**Knowledge probed from large language models.** While LLM-based visual descriptors have been explored for fine-grained and cross-geography object recognition (Pratt et al., 2023; Menon & Vondrick, 2023; Saha et al., 2024; Buettner et al., 2024), this is the first work to use descriptors for evaluating cultural competence in T2I models.

## 3 METHODOLOGY

Evaluating cultural faithfulness requires more than image-text alignment. A reliable metric must reward correct cultural elements (e.g., interactions, objects, attire) while penalizing hallucinated or exaggerated ones. However, Fig. 5a shows existing VLM-based metrics do not correlate negatively with hallucination and exaggeration. This limitation motivates a structured framework utilizing external cultural concepts to evaluate images.

We introduce AHEaD, a descriptor-based framework designed for culturally faithful image generation. First, each activity is annotated automatically with a comprehensive set of culturally-grounded reference descriptors ($\mathcal{D}^{ref}$). Then, predicted descriptors ($\mathcal{D}^{pred}$) are extracted from images. Finally, AHEaD metrics are computed by comparing predicted and reference descriptors, yielding descriptor-based feedback as illustrated in Fig. 2. Sec. 3.1 describes reference descriptor generation and Sec. 3.2 defines AHEaD metrics.

### 3.1 REFERENCE DESCRIPTOR GENERATION

We represent each activity-country pair as a set of cultural descriptors, $\mathcal{D}^{ref}$, encoding expected visual elements across five dimensions: *background* (e.g., Eiffel tower, geometric patterns), *attire* (e.g., traditional vs. modern clothing), *objects*, *actions/interactions* (e.g., greeting with a bow), and *spatial layout* (e.g., dancers in a circle). These descriptors are LLM-generated to establish a ref-

erence for evaluation independent of images. Concretely, we construct descriptors using Proposer-Refiner, a two-stage method inspired by self-consistency prompting (Wang et al., 2023). First, the **Proposer** leverages multiple LLMs to independently generate up to 10 mutually exclusive descriptors per dimension. Using multiple LLMs increases coverage while mitigating model-specific bias while capturing diverse cultural variants. Second, the **Refiner** filters these candidates to remove duplicates and errors, enhancing precision (example in Fig. 2). Tab. 4, the Proposer-Refiner framework significantly improves descriptor quality over single-stage generation.

## 3.2 AHEAD EVALUATION METRICS

We design descriptor-based metrics along two complementary axes, *faithfulness* and *diversity*. A faithful T2I model should (i) cover the expected elements (Alignment), (ii) avoid irrelevant elements to the activity or culture (Hallucination), (iii) refrain from over-exaggerating cultural elements (Exaggeration). We also measure variety across generations (Diversity).

For each activity $a$ and region $r$ (country), we generate $N$ images $\{I_n\}_{n=1}^N$ using the prompt $T_{r,a}$. Rather than using an MLLM to score images directly, we employ it to parse visual content into fine-grained descriptors $\mathcal{D}^{pred}(I_n)$. These are aggregated into a set $\mathcal{D}_{r,a}^{pred} = \bigcup_{n=1}^N \mathcal{D}^{pred}(I_n)$ across the same five cultural dimensions as $\mathcal{D}_{r,a}^{ref}$.

To build AHEaD metrics for $x_{r,a} = (\{I_n\}_{n=1}^N, \mathcal{D}_{r,a}^{pred}, \mathcal{D}_{r,a}^{ref})$, we construct a complete bipartite graph between predicted and reference descriptors. In this graph, predicted descriptors $\hat{d}_i \in \mathcal{D}_{r,a}^{pred}$ and reference descriptors $d_j \in \mathcal{D}_{r,a}^{ref}$ serve as nodes. Each edge encodes the semantic similarity $s_{i,j} = \text{sim}(\hat{d}_i, d_j)$ between descriptors, as measured by sentence embeddings.

**Alignment**. Alignment measures coverage of expected cultural elements. For each reference descriptor $d_j$, we identify its best match predicted descriptor via maximum edge similarity. Alignment is the fraction of reference descriptors whose best match exceeds threshold $\tau$:

$$\text{ALIGN}(x_{r,a}) = \frac{1}{|\mathcal{D}_{r,a}^{ref}|} \sum_{d_j \in \mathcal{D}_{r,a}^{ref}} \mathbb{1}\left[\max_i s_{i,j} > \tau\right] \tag{1}$$

where $\mathbb{1}[\cdot]$ is the indicator function and $\tau$ is calibrated according to real images (Sec. A.3). The final score is the average of scores across five cultural dimensions (e.g., spatial, attire, objects, etc).

**Hallucination.** While Alignment measures coverage, it does not capture extraneous or culturally incorrect elements. We address this via HAL, defined as the fraction of predicted descriptors $\hat{d}_i$ that lack a corresponding match in the reference set:

$$\text{HAL}(x_{r,a}) = \frac{1}{|\mathcal{D}_{r,a}^{pred}|} \sum_{\hat{d}_i \in \mathcal{D}_{r,a}^{pred}} \mathbb{1}\left[\max_j s_{i,j} \leq \tau\right] \tag{2}$$

In practice, we measure this for each cultural dimension separately and report the average.

**Exaggeration.** A faithful generation must also avoid over-emphasizing stereotypical elements. We measure exaggeration by comparing the intensity of stereotypical elements in generated images against real images scraped from web (details in Sec. 4.1). First, inspired by D'Incà et al. (2024), we use an LLM to propose a set of stereotypical candidates $S_r$ for region $r$. Let $f(I, d_k)$ denote the ITA score between an image $I$ and a candidate descriptor $d_k \in S_r$. We define the baseline reference score as the average ITA score over $n_{gt}$ real images:

$$\bar{f}_{gt}(d_k) = \frac{1}{n_{gt}} \sum_{m=1}^{n_{gt}} f(I_{gt}^m, d_k) \tag{3}$$

For an instance $x_{r,a}$, the exaggeration score is the average maximum positive deviation from this baseline across $N$ generations:

$$\text{EXAG}(x_{r,a}) = \frac{1}{N} \sum_{n=1}^{N} \max_{d_k \in S_r} \left[ \max\left(0, f(I_n, d_k) - \bar{f}_{gt}(d_k)\right) \right] \tag{4}$$

**Faithfulness.** Cultural faithfulness is defined as a composite score:

$$\text{FAITH}(x_{r,a}) = g\big(\text{ALIGN}, 1 - \text{HAL}, 1 - \text{EXAG}\big) \tag{5}$$

where $g(\cdot)$ is the arithmetic mean. Additionally, AHEaD provides interpretable descriptor-based feedback identifying aligned, missing, and exaggerated elements which can enable descriptor-guided image editing (see Fig. 8 for a preliminary result).

**Descriptor Diversity.** Beyond faithfulness, we measure the variety of cultural elements produced across N generations for $x_{r,a}$ using normalized entropy:

$$\text{DDIV}(x_{r,a}) = \frac{-1}{\log |\mathcal{D}_{r,a}^{ref}|} \sum_{d \in \mathcal{D}_{r,a}^{ref}, q(d) > 0} q(d) \log q(d) \tag{6}$$

$q(d)$ is normalized frequency of reference descriptor $d$ appearing across $N$ images ($\sum_d q(d) = 1$).

**Semantic Diversity.** We define semantic diversity as the marginal utility in descriptor coverage provided by $N$ images over a single generation:

$$\text{SDIV}(x_{r,a}) = \text{ALIGN}_N(x_{r,a}) - \mathbb{E}[\text{ALIGN}_1(x_{r,a})] \tag{7}$$

where $\text{ALIGN}_N$ is alignment over $N$ images and $\mathbb{E}[\text{ALIGN}_1]$ is average single-image alignment.

## 4 EXPERIMENTAL SETUP

### 4.1 CULTIVATE BENCHMARK

Constructing cross-cultural benchmarks with local activities is challenging as it requires expert regional knowledge. To obtain this knowledge systematically, we parse existing knowledge bases, CulturalAtlas[1] and Wikipedia, with GPT-4o to extract non-overlapping activities per country. These sources are complementary with CulturalAtlas documenting cultural practices (e.g., greetings, religious customs, etiquette), and Wikipedia providing activity lists (e.g., games, celebrations). CUL-TIVate spans 16 countries with 576 activities across 9 categories, generating 19,000+ images from 6 T2I models with comprehensive ground-truth descriptor annotations.

**Activities.** We consider 9 activity categories falling into three types: (1) *multi-variant* (dances, games, religious practices, greetings, celebrations) with multiple activities per country (e.g., different traditional dances), (2) *setting-based* (eating, concerts) with activities varying by context (home/restaurant, indoor/outdoor), and (3) *single-variant* (weddings, funerals) with one activity per country.

**Countries.** We select 16 countries spanning all socio-cultural regions in CulturalAtlas. Following UN classification[2], countries are divided into Global North (USA, Spain, Italy, Germany, France) and Global South (Iran, Turkey, China, India, Indonesia, Philippines, Nepal, Nigeria, South Africa, Brazil, Mexico).

**Image Generation.** For each prompt, we generate images using the template *"A photorealistic photo of {activity} in {country}."* We evaluate 6 recent T2I models: 3 public (Stable Diffusion 3.5 (Esser et al., 2024), FLUX (BlackForestLabs, 2024), Qwen-Image (Wu et al., 2025)[3]) and 3 proprietary (DALL·E 3 (Betker et al., 2023), GPT-Image-1 (OpenAI, 2025), Gemini 2.5 Flash Image (i.e. Nano Banana) (Google, 2025)). For public models, we generate 10 images per prompt with random seeds $42 + i$ for $i$-th image. For proprietary models, we generate 1 image due to the cost.

---

[1]https://culturalatlas.sbs.com.au/

[2]https://unctadstat.unctad.org/EN/Classifications/DimCountries_All_Hierarchy.pdf

[3]We used the distilled model: https://github.com/ModelTC/Qwen-Image-Lightning

**Reference data.** We adopt two complementary strategies for identifying what images of activities in a region (country) should portray: (1) *Visual Descriptors*: We extend prior usage of LLMs for object descriptors (Menon & Vondrick, 2023) to activities. For each prompt, we generate up to 10 descriptors per 5 dimensions, capturing diverse valid variants of each activity (details in Sec. 3.1); (2) *Real Images:* We collect 20 candidate images per prompt via Google search (10 using the English prompt, 10 using its translation into the language of the respective country), totaling $\sim$12k images. We then apply CLIPScore (Hessel et al., 2021) filtering and retain the top five (total of $\sim$3k) as representative real references which we use in our EXAGgeration metric. Real images also serve for calibration and hyperparameter tuning (e.g., $\tau$ in Eq. 1).

## 4.2 HUMAN EVALUATION SETUP

We conduct a human study to validate our metrics for Faithfulness (i.e., Alignment, Hallucination, and Exaggeration) and realism (whether image looks realistic). We adopt Prolific[4] as our platform.

**Study Design.** Our evaluation spans 11 countries across 5 CulturalAtlas regions (Middle East, America, Europe, Africa, Asia) and both Global North and South. We selected 1–2 activities per category, ensuring coverage of all activity groups per country. We assessed three public T2I models and real images (for 3 countries), totaling 398 forms and 796 annotations (two annotators per form).

**Annotations.** We collected 3 ground-truth (GT) labels using a 5-point Likert scale: ***GT-FAITH*** (main gold standard) measures overall faithfulness according to annotators—*How well does this image show {activity} in your country?*; ***GT-EXAG*** measures image exaggeration—*How exaggerated is the image?* ; ***GT-HAL*** measures hallucinations—*How incorrect is the image? (activity/culture)*.

We also evaluate reference descriptor quality through human evaluation. We measure **precision** by having annotators mark each descriptor as correct or incorrect; 90% of descriptors were marked correct (Tab. 11). It is infeasible to compute **recall** because no ground-truth descriptor set exists. Instead, we estimate recall through two measures: (1) average coverage rating of 4.5/5, and (2) only 26 of 378 annotators reported missing descriptors. These results demonstrate high precision and comprehensive coverage. We utilize two complementary measures: (1) average descriptor quality rating (result: 4.5/5), and (2) proportion of annotators reporting at least one missing descriptor (result: 26/378). These results demonstrate high precision and comprehensive coverage.

**Quality Control.** We recruit annotators matching each country's nationality (verified by Prolific) at \$8/hour compensation. To ensure reliability, we implement multiple quality control measures, such as attention checks (e.g., selecting a pre-mentioned number), repeated questions to test consistency, and required free-text rationales describing the errors in the image. We also conducted direct discussions with annotators when facing inconsistent scores and explanations.

**Correlation Metric & Inter-rater Agreement.** We use Spearman's rank correlation to measure how well our proposed metrics align with human judgments. Spearman's $\rho$ evaluates the strength of monotonic relationships between ranked variables, where values near 1/-1 indicates a strong positive/negative correlation. Following prior work (Kannen et al., 2024), we measure inter-rater agreement using Krippendorff's Alpha (Krippendorff, 2018), which is well-suited for ordinal Likert scales. We compute agreement separately for each country. Agreement is moderate and varies by country, consistent with prior cross-cultural studies (Nayak et al., 2025; Kannen et al., 2024), reflecting cultural evaluation's inherent subjectivity. Our agreement levels are comparable to or exceed their maximum per-country values (Appendix A.4).

## 4.3 IMPLEMENTATION DETAILS AND BASELINES

**Implementation Details.** For Proposer–Refiner, we use Gemini 2.5 Flash (Comanici et al., 2025) and GPT-4o (Hurst et al., 2024). GPT-4o is used as the refiner due to its sota cultural understanding (Chiu et al., 2025a).We use InternVL3-14B (Zhu et al., 2025) and Qwen2.5-VL-7B-Instruct (Bai et al., 2025) as MLLMs in AHEaD, and compute sentence embeddings with all-MiniLM-L6-v2. $\tau$ is calibrated on real images: 0.52 (InternVL3), 0.67 (Qwen2.5-VL). Details are in Sec. A.4.

**Baselines.** We compare AHEaD against ITA metrics, such as CLIPScore (Hessel et al., 2021), ImageReward (Xu et al., 2023), VIEScore (Ku et al., 2024), VQAScore (Lin et al., 2024),

---

[4]https://www.prolific.com/

| Model | Region | N=1 | | | | N=10 | |
|-------|--------|---------|--------|--------|---------|---------|---------|
| | | **ALIGN ↑** | **HAL ↓** | **EXAG ↓** | **FAITH ↑** | **DDIV ↑** | **SDIV ↑** |
| SD-3.5-medium | GN | **0.31** ±0.01 | **0.55** ±0.02 | **0.05** ±0.02 | **0.57** ±- | **0.68** ±0.03 | **0.33** ±- |
| | GS | 0.26 ±0.03 | 0.61 ±0.03 | 0.08 ±0.04 | 0.52 ±- | 0.62 ±0.04 | 0.32 ±- |
| FLUX.1-dev | GN | **0.30** ±0.02 | **0.56** ±0.03 | **0.04** ±0.01 | **0.57** ±- | **0.66** ±0.03 | **0.32** ±- |
| | GS | 0.25 ±0.03 | 0.63 ±0.04 | 0.06 ±0.02 | 0.52 ±- | 0.60 ±0.04 | 0.30 ±- |
| Qwen-Image | GN | **0.36** ±0.02 | **0.51** ±0.02 | **0.06** ±0.01 | **0.60** ±- | **0.68** ±0.02 | 0.28 ±- |
| | GS | 0.30 ±0.03 | 0.56 ±0.04 | 0.10 ±0.03 | 0.55 ±- | 0.63 ±0.04 | **0.29** ±- |
| DALL-E 3 | GN | **0.36** ±0.01 | **0.50** ±0.01 | **0.10** ±0.03 | **0.59** ±- | - | - |
| | GS | 0.32 ±0.03 | 0.54 ±0.032 | 0.12 ±0.04 | 0.55 ±- | - | - |
| GPT-Image-1 | GN | **0.36** ±0.01 | **0.49** ±0.01 | **0.06** ±0.01 | **0.61** ±- | - | - |
| | GS | 0.30 ±0.03 | 0.55 ±0.03 | 0.07 ±0.02 | 0.56 ±- | - | - |
| Gemini 2.5 Flash Image | GN | **0.40** ±0.01 | **0.46** ±0.01 | **0.10** ±0.03 | **0.61** ±- | - | - |
| | GS | 0.35 ±0.03 | 0.50 ±0.03 | 0.12 ±0.3 | 0.57 ±- | - | - |

Table 1: **T2I models consistently generate more faithful images on GN countries.** $N$ is number of images per prompt. Best values per model (GS/GN) is **bolded**. EXAG values are small because it measures relative alignment of synthetic image to real images.

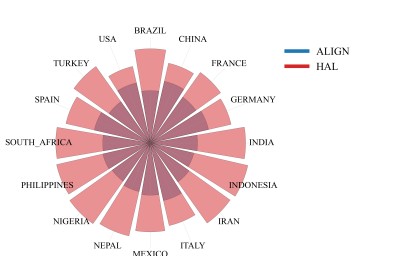

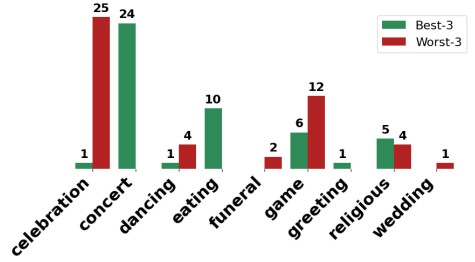

(a) Models consistently score better (higher on ALIGNment, lower on HALlucation) for GN. Interestingly, they perform strongly on China

(b) Frequency of activities appearing among the best-3 (green) and worst-3 (red) across countries.

Figure 3: Analysis of performance by country (left) and activity (right).

PickScore (Kirstain et al., 2023) and existing cultural metrics CURE (Rege et al., 2025). For fair comparison, we include MLLM-as-Judge baselines with the same backbone. Additional details in Sec. A.4.

## 5 RESULTS

### 5.1 HOW DO DIFFERENT T2I MODELS PERFORM FOR DIFFERENT COUNTRIES?

**T2I models consistently generate more faithful images for Global North countries.** Tab. 1 shows a consistent bias against GS, with all models performing better on GN. For example, Qwen-Image achieves higher FAITH (0.60 vs 0.55), higher ALIGN (0.36 vs 0.30), and lower HAL (0.51 vs 0.56) on GN compared to GS. This pattern holds across models where lower ALIGN, higher HAL/EXAG, and lower DDIV/SDIV on GS indicate models make more errors, generate more exaggerated content, and exhibit less diversity for Global South countries. Fig. 3a shows per-country trends.

**T2I systems struggle most with culturally grounded activities.** Fig. 3b shows the frequency of each activity appearing in the top-3 and bottom-3 performing tiers across 16 countries. Performance is highest for universal activities (e.g., concerts, eating) and lowest for culturally grounded ones (e.g., celebrations), indicating a gap in modeling specific cultural contexts.

| Backbone | Method | GT-FAITH | | |
|---|---|---|---|---|
| | | **GS** | **GN** | **All** |
| PickScore (Kirstain et al., 2023) | | **0.20** | -0.02 | **0.15** |
| ImageReward (Xu et al., 2023) | | -0.03 | -0.13 | -0.08 |
| CLIPScore (Hessel et al., 2021) | I–T Alignment | 0.08 | -0.01 | 0.04 |
| VQAScore (Lin et al., 2024) | | 0.15 | **0.16** | 0.14 |
| CuRe (Rege et al., 2025) | | 0.13 | 0.08 | 0.10 |
| Qwen2.5–VL (Bai et al., 2025) | MLLM | 0.13 | 0.08 | 0.10 |
| | **FAITH (Ours)** | **0.42** (+0.29) | **0.38** (+0.30) | **0.42** (+0.32) |
| InternVL3 (Zhu et al., 2025) | MLLM | 0.19 | 0.18 | 0.20 |
| | **FAITH (Ours)** | **0.46** (+0.27) | **0.47** (+0.29) | **0.47** (+0.27) |
| GPT-4o | MLLM | **0.49** | **0.46** | **0.48** |
| | VIEScore (Ku et al., 2024) | 0.37 | 0.27 | 0.35 |
| – | Human | 0.59 | 0.57 | 0.58 |

Table 2: **FAITH achieves significantly higher correlation with human judgment on faithfulness compared to ITA and MLLM-as-a-Judge baselines.** FAITH shows consistent performance across backbones and achieves comparable results to GPT-4o despite using much weaker backbone. Best values per each section are **bolded**. Values in parentheses show improvement over MLLM baseline with the same backbone. Human–human correlation is provided for reference.

| Backbone | MLLM Baseline | | | ALIGN | | | ALIGN+HAL | | | FAITH | | |
|---|---|---|---|---|---|---|---|---|---|---|---|---|
| | GS | GN | All | GS | GN | All | GS | GN | All | GS | GN | All |
| Qwen2.5-VL (Bai et al., 2025) | 0.13 | 0.08 | 0.10 | 0.41 | 0.32 | 0.39 | 0.37 | 0.37 | 0.39 | **0.42** | **0.38** | **0.42** |
| InternVL3 Zhu et al. (2025) | 0.19 | 0.18 | 0.20 | 0.40 | 0.40 | 0.41 | 0.42 | 0.46 | 0.44 | **0.46** | **0.47** | **0.47** |

Table 3: **Necessity of composite metrics for *faithfulness*.** Spearman correlation with GT-FAITH across regions. Our composite metric (FAITH) significantly outperforms individual components, confirming that alignment alone is insufficient and is complemented by HAL and EXAG. Correlations are statistically significant ($p \leq 0.0001$)

## 5.2 WHAT METRICS ARE EFFECTIVE FOR CULTURAL FAITHFULLNESS?

**Image-Text Alignment metrics are ineffective for cultural understanding.** Tab. 2 shows ITAs' correlation with humans scores are below 0.15 (e.g., ImageReward: -0.03/-0.13/-0.08 for GS/GN/all). MLLM-as-judge baselines, which prompt MLLMs with the same questions given to annotators, correlate better but still substantially lag behind FAITH (e.g., Qwen2.5-VL: 0.10 vs FAITH: 0.42 and InternVL3: 0.20 vs FAITH: 0.47 on all).

**EXAGgeration and HALlucination complement ALIGNment.** Tab. 3 demonstrates that combining ALIGN and HAL improves correlation with human faithfulness and best results is achieved when all three metrics are combined (i.e. FAITH). These results indicate that effective cultural faithfulness metrics must penalize exaggeration and hallucination rather than rely on alignment alone.

**Ablations.** Tab. 4 compares proposer-refiner against proposer-only for descriptor generation, showing the two-stage approach improves correlation with human faithfulness judgments. Tab. 5 ablates the threshold parameter $\tau$ (Section 3.2) used to determine whether reference and predicted descriptors match. We test values at the 25th, 50th, and 75th percentiles, finding the 75th percentile performs best.

## 5.3 WHAT ASPECTS OF THE ACTIVITIES ARE DEPICTED BEST/WORST BY T2I MODELS?

Fig. 4 shows performance by region (country) across each of five descriptor dimensions (see Sec. 3). The best-performing country varies by dimension, but the USA, China, and Germany consistently rank among the best. South Africa, Nigeria, and India generally fall in the lower half, except for Interaction (South Africa and Nigeria) and Spatial (India).

| Ref. Desc. Generator | LLM (Proposer/ Refiner) | Spearman | Kendall |
|---|---|---|---|
| Proposer | GPT-4o / – | 0.28 | 0.20 |
| Proposer | Gemini 2.5-Flash / – | 0.30 | 0.22 |
| Proposer-Refiner | GPT-4o + Gemini 2.5-Flash / GPT-4o | **0.33** | **0.24** |

| $\tau$ | Spearman | Kendall |
|---|---|---|
| 0.29 | 0.21 | 0.15 |
| 0.39 | 0.27 | 0.20 |
| 0.52 | **0.33** | **0.24** |

Table 4: **Proposer–Refiner improves descriptor quality**.

Table 5: **Thresh. ($\tau$) ablation**.

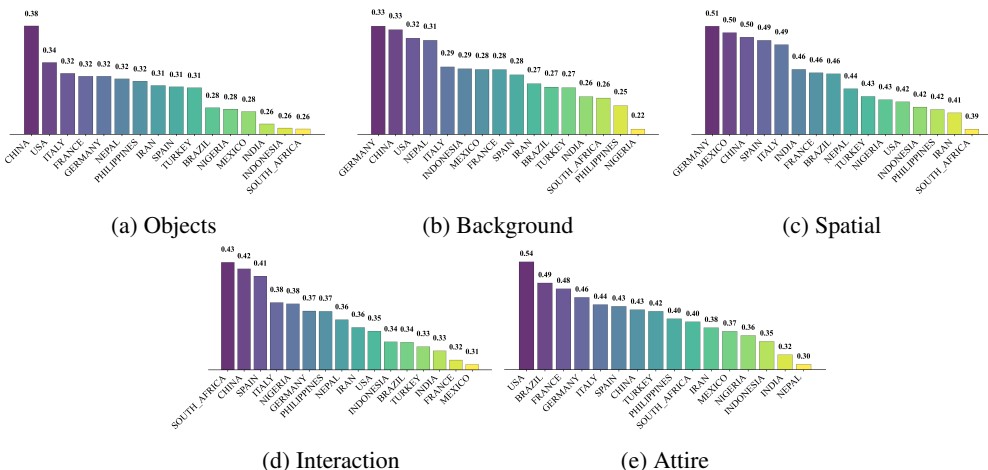

(a) Objects  (b) Background  (c) Spatial

(d) Interaction  (e) Attire

Figure 4: Country alignment ranked using each of the five descriptor dimensions.

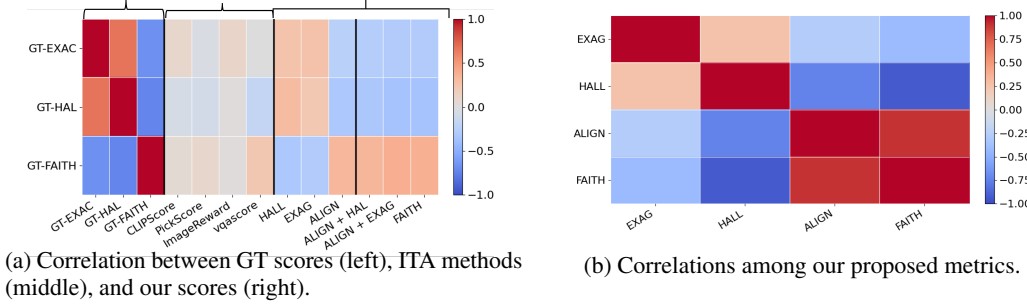

(a) Correlation between GT scores (left), ITA methods (middle), and our scores (right).

(b) Correlations among our proposed metrics.

Figure 5: Effective *faithfulness* metrics must negatively correlate with exaggeration/hallucination.

### 5.4 How do the metrics relate to each other?

To improve the performance of T2I models, a user might want to know how improving upon one metric will affect others. We aim to answer this question by computing correlations between the metrics, shown in Fig. 5b. We see that alignment is negatively correlated with both exaggeration and hallucination. The same trend is observed using human scores; see Fig. 5a which also demonstrates visually the much stronger alignment of our metrics with human scores.

## 6 Conclusion

We developed a framework for evaluating generation of images of social activities in different countries. We propose a suite of metrics that can be computed without human involvement, yet show much higher agreement with human assessment than prior metrics. Using our framework, we conduct analysis on sixteen countries and six text-to-image models. We show performance on Global North countries exceeds that of Global South. and demonstrate specific failure modes using our descriptor dimensions. We hope our work equips future researchers with the tools to scalably improve and test performance on this task which has broad applicability, e.g., in the entertainment industry.

## 7 ACKNOWLEDGMENT

This work is supported by NSF Grant No. 2329992. We gratefully acknowledge the support of those who contributed to the human evaluation. We also thank Aysan Aghazadeh and Christopher Achkar for their valuable comments and help throughout the project.

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

# A    APPENDIX

## A.1    USAGE OF AI

In this section we elaborate on LLM usage in this study. LLMs were used throughout this research as writing assistants, for text polishing, and for literature review through LLM agents and available tools. AI coding assistants[5] were used to assist with programming. However, LLMs were not used blindly and served only as assistants to improve accuracy and efficiency. This paper introduces a benchmark on social activities. As described in the main paper, LLMs (GPT-4o) were utilized to parse online knowledge bases (CulturalAtlas and Wikipedia) to identify activities across countries. Furthermore, the descriptor-based metrics rely on LLM-generated descriptors. However, a proposer-refiner approach was incorporated to improve quality, and descriptors were evaluated through human evaluation (see Table 11).

## A.2    LIMITATIONS

**Cultural Bias in LLMs**. AHEaD uses LLM-generated descriptors as reference points for measuring the cultural competence of T2I models. Since LLMs are trained on web text, we acknowledge that they may encode biases toward Western societies. To mitigate this, we adopt a Proposer–Refiner strategy, which improves descriptor quality and increases agreement with human ground-truth scores. Human evaluation showed 90%. Compared to common alternatives, such as human surveys or real images, our approach is scalable and less costly. Real images collected from the web are themselves biased, while surveys are subjective and expensive. Unlike VLM-based image–text alignment methods or raw image references, our **descriptors are explainable and allow direct inspection of model errors**, rather than being opaque scores.

## A.3    CALIBRATION OF THRESHOLD

We propose ALIGN and HAL to measure *how well images cover expected activity/cultural cues* and which *visual elements are incorrect*. Since these metrics are ratio-based, we must set a similarity threshold $\tau$ to decide whether a descriptor counts as a *hit* (aligned) or *miss* (hallucinated).

We calibrate $\tau$ using real reference images rather than synthetic generations to avoid leakage, since synthetic data may reflect biases of the very T2I models under evaluation. Real images, while noisy, contain culturally faithful content without "wrong" or "exaggerated" elements, making them suitable for calibration. Concretely, we compute descriptor–descriptor similarities between LLM-provided ground-truth descriptors and MLLM-extracted descriptors from real images, then consider candidate thresholds at the lower quartile (Q1), median, and upper quartile (Q3). As shown in Fig.6, Q3 offers the best trade-off by reducing false positives while maintaining recall. Table5 further confirms that Q3 yields the most robust alignment scores across regions.

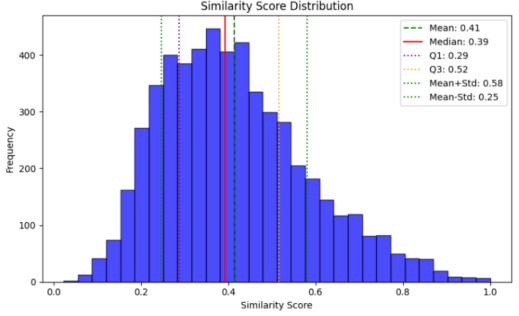

Figure 6: **Threshold $\tau$ calibration for ALIGN**

---

[5]https://cursor.com/

## A.4 Implementation Details and Baselines

**Evaluation baselines**. The goal of this paper is to evaluate the cultural faithfulness competence (GT-ALIGN) of T2I models, where automated evaluation methods remain extremely limited. Existing works rely heavily on human annotations (Kannen et al., 2024; Nayak et al., 2025; Basu et al., 2023), while a few recent approaches (Khanuja et al., 2024; Rege et al., 2025) approximate cultural faithfulness using image–text similarity. Accordingly, we compare against commonly used and state-of-the-art ITA metrics, including CLIPScore (Hessel et al., 2021), VQAScore (Lin et al., 2024) with "CLIP-FlanT5-xxl" (the strongest publicly available ITA setup), PickScore (Kirstain et al., 2023), and ImageReward (Xu et al., 2023). Following prior ITA practice, we use each model's generation prompt–"A photorealistic image of activity in country"–as the reference for evaluation. We also benchmark against CuRe (Rege et al., 2025), the only metric explicitly designed for cultural faithfulness. For fair comparison, we adopt CuRe's recommended SigLIP2 (Tschannen et al., 2025) configuration and compute mean image–text similarity using the prompts "An image of activity" and "An image from country," omitting their parent-category prompt since this information is already embedded in our activity descriptions (e.g., "people playing tag game").

Across all settings, we find that ITA methods and CuRe exhibit weak correlation with human cultural judgments, whereas our proposed metrics achieve substantially higher and more stable agreement across different MLLM backbones (InternVL3 and QwenVL2.5). We attribute the limitations of existing VLM-based ITA methods to: (1) bag-of-words behavior that misses compositional cultural nuance (Yuksekgonul et al., 2023), (2) reliance on Western-centric training data that introduces cultural biases, and (3) inability to distinguish authentic cultural representation from stereotypical exaggeration. For instance, CLIPScore rewards images containing literal elephants for the "elephant ant man" game-an Indonesian rock–paper-scissors variant– due to keyword matching rather than cultural understanding. To address these issues, AHEAD uses externally generated cultural descriptors instead of VLM embeddings, enabling interpretable evaluation of ALIGN, HAL, and EXAG that aligns more faithfully with human cultural judgment. This is the first work to evaluate cultural HAL and EXAG, and we study both descriptor–descriptor methods (Sec. 3.2) and MLLM-as-Judge baselines using InternVL3 and QwenVL2.5, which answer the same cultural assessment questions posed to human annotators (full prompts in Appendix A.6).

**Implementation Details** We first use GPT4-o and Gemini 2.5 Flash (best LLMs in cultural understanding (Chiu et al., 2025b)) offline once to in the data annotation phase to produce "reference LLM descriptors", these are used as noisy reference to evaluate cultural faithfulness. To minimize the LLM-bias we developed proposer-refiner to combine descriptors of different LLMs which is refined by removing duplicate and incorrect descriptors (results in Table 4). We set the temperature to 0.2 for proposers and 0.1 for the refiner. AHEaD uses an MLLM to extract descriptors, we mainly use InternVL3 ("InternVL3-14B") as MLLM in our pipeline and also test our pipeline with QwenVL2.5("QwenVL2.5-7B"). We set temperature 0 for MLLMs to ensure high precision and reproducibility, and use `all-MiniLM-L6-v2` as the sentence embedding model for similarity computation. All experiments run on a single L40S GPU.

**Inter-Rater Agreement.** We consider "GT-ALIGN" for inter-rater agreement as the main goal of this work is to measure cultural faithfulness as GT-EXAG/GT-HAL are more subjective. To assess the reliability of our human annotations, we compute country-level agreement scores for the cultural relevance ratings. Each image is annotated by two independent raters who are originally from the corresponding country. Across the eleven countries in our study, Krippendorff's Alpha (Krippendorff, 2018) ranges from $0.15$ to $0.62$. We also compute Cohen's Kappa (McHugh, 2012) between the two annotator groups and observe a mean value of $0.50$. These agreement levels are consistent with previously reported values for cross-cultural image evaluation. CulturalFrames (Nayak et al., 2025) reports country-level Alpha values between $0.24$ and $0.42$, and CUBE (Kannen et al., 2024) reports values between $0.09$ and $0.58$. Our scores are therefore comparable to prior work and also achieve a higher maximum value, which indicates that our annotation protocol yields reliable judgments.

We observe variation across countries, with a standard deviation of $0.13$ for Krippendorff's Alpha. Such variation is expected because cultural faithfulness assessments are subjective and depend strongly on cultural and geographic context. Interestingly, the average agreement among Global North countries is $0.28$, which is lower than the Global South average of $0.35$, even though text-to-image models tend to perform better on Global North regions. We hypothesize that higher-quality

| Method | Model | Flux1 | Qwen-Image | SD3.5 | Avg. |
|---|---|---|---|---|---|
| Image-Text Alignment | VQAScore | 0.14 | -0.02 | 0.14 | 0.09 |
| | PickScore | 0.06 | -0.05 | 0.03 | 0.01 |
| | ImageReward | -0.09 | -0.24 | -0.19 | -0.17 |
| | CLIPScore | 0.04 | -0.28 | 0.02 | -0.05 |
| | CuRe | 0.17 | 0.10 | -0.01 | 0.09 |
| MLLM | GPT-4o | **0.53** | 0.27 | **0.48** | **0.43** |
| | InternVL3 | 0.38 | -0.14 | 0.14 | 0.13 |
| | QwenVL2.5 | 0.12 | 0.02 | 0.11 | 0.09 |
| ALIGN | InternVL3 | 0.51 | **0.30** | 0.31 | 0.38 |
| Human | – | 0.65 | 0.40 | 0.55 | 0.55 |

Table 6: **Per-model correlation with GT-FAITH.** ALIGN with InternVL3 backbone significantly outperforms all ITA metrics and InternVL3-as-judge baseline across T2I models, achieving performance comparable to GPT-4o-as-judge.

| Method | Backbone | GT-HAL↑ | | | GT-FAITH↓ | | |
|---|---|---|---|---|---|---|---|
| | | GS | GN | overall | GS | GN | overall |
| MLLM | InternVL3 | 0.22 | 0.24 | 0.23 | -0.20 | -0.24 | -0.21 |
| | QwenVL2.5 | 0.29 | 0.30 | 0.29 | -0.31 | -0.36 | -0.33 |
| HAL | InternVL3 | **0.31** | 0.39 | 0.35 | **-0.39** | **-0.44** | **-0.41** |
| | QwenVL2.5 | 0.30 | **0.42** | **0.36** | -0.33 | -0.35 | -0.36 |
| Human | – | 0.40 | 0.38 | 0.39 | - | - | - |

Table 7: **Correlation with humans on Hallucination.** Our Hallucination metric achieves the highest correlation with human ground truth scores compared to existing MLLM-based approaches, including InternVL which serves as the backbone for MLLM descriptor extraction. Best scores per column are **bolded**.

outputs may cause annotators to focus more on aspects unrelated to cultural content, such as image quality or visual artifacts, or to rely more heavily on subjective interpretations.

## A.5 Additional Results on AHEaD

**Per-T2I Alignment Performance**. Table 6 compares correlation with human faithfulness judgments across T2I models. ITA metrics perform poorly, with most showing near-zero or negative correlation (e.g., CLIPScore averages -0.05, ImageReward -0.17). MLLM-as-judge baselines achieve higher correlation, with GPT-4o reaching 0.43 average. Our ALIGN metric with InternVL3 backbone achieves 0.38 average correlation, significantly outperforming all ITA methods and InternVL3-as-judge (0.13), while approaching GPT-4o performance despite using a weaker model. Human-human agreement is moderate (0.55).

**HAL can effectively detect hallucinations.** Table 7 shows our HAL metric achieves highest correlation with GT-HAL and lowest with GT-FAITH, outperforming MLLM baselines. For example, although we use InternVL3 to extract descriptors, our HAL outperforms InternVL3-as-judge by 11% on GT-FAITH correlation. Notably, HAL exhibits strong negative correlation with GT-FAITH, validating that hallucination degrades faithfulness. Table 8 shows per-model results where HAL significantly outperforms MLLM baselines across all T2I systems (e.g., +30%/+20% on Qwen-Image for InternVL3/Qwen2.5-VL backbones).

**EXAG can effectively detect exaggeration.** We are the first to measure exaggeration for cultural faithfulness evaluation. We explore two approaches: ITA-based using exaggeration candidates (Section 3.2) and MLLM-based prompting models to detect exaggeration. Additionally Tab. 12 illustrates top-3 descriptor examples.

| Method | Backbone | GT-HAL↑ | | | | GT-FAITH↓ | | | |
|---|---|---|---|---|---|---|---|---|---|
| | | Flux1 | Qwen-Image | SD3.5 | Avg. | Flux1 | Qwen-Image | SD3.5 | Avg. |
| MLLM | InternVL3 | 0.32 | 0.07 | 0.20 | 0.20 | -0.30 | -0.10 | -0.18 | -0.18 |
| | QwenVL2.5 | **0.38** | 0.11 | **0.37** | 0.26 | -0.39 | -0.19 | -0.31 | -0.30 |
| HAL | InternVL3 | 0.36 | **0.37** | 0.24 | 0.32 | **-0.51** | **-0.33** | -0.30 | **-0.38** |
| | QwenVL2.5 | 0.35 | 0.31 | 0.32 | **0.33** | -0.38 | -0.21 | **-0.37** | -0.32 |
| Human | – | 0.38 | 0.34 | 0.40 | 0.37 | - | - | - | - |

Table 8: **Hallucination Per T2I**. Spearman Correlation.

| Method | Backbone | GT-EXAG↑ | | | GT-FAITH↓ | | |
|---|---|---|---|---|---|---|---|
| | | GS | GN | All | GS | GN | All |
| EXAG (MLLM) | InternVL3 | 0.24 | 0.26 | 0.25 | -0.24 | -0.21 | -0.25 |
| | QwenVL2.5 | 0.34 | **0.33** | **0.34** | **-0.31** | **-0.25** | **-0.30** |
| EXAG (ITA) | VQAScore | **0.36** | 0.16 | 0.29 | -0.27 | -0.14 | -0.22 |
| Human | – | 0.31 | 0.39 | 0.34 | - | - | - |

Table 9: **Correlation with humans on Exaggeration.** Best scores per-column are bolded. We explore two approaches: EXAG(MLLM) use MLLM for predicting exaggeration, while EXAG(ITA) uses VQAScore and exaggerated candidates from Sec. 3.2.

Table 9 compares ITA-based (VQAScore) versus MLLM-based approaches. While MLLM-based methods show stronger correlation with GT-EXAG, we adopt the ITA-based approach in our framework because it provides interpretable descriptor-level feedback. Our framework supports both approaches, allowing users to choose based on their needs for interpretability versus performance.

Table 10 compares three descriptor sources for ITA-based EXAG: (1) LLM-generated stereotype candidates, (2) reference descriptors ($\mathcal{D}^{ref}$), and (3) MLLM-extracted descriptors from real images. Stereotype candidates achieve significantly better correlation with GT-EXAG (0.183 average) compared to reference descriptors (-0.251) and MLLM descriptors (-0.083). This demonstrates that general descriptors fail to capture exaggeration, as they include non-stereotypical correct elements. Only stereotype-specific candidates effectively measure over-representation.

**Correlation among metrics**. Figure 7 examines relationships between our metrics and human judgments. Both humans and our automatic metrics show strong negative correlation between ALIGN and HAL (-0.73/-0.74) and between ALIGN and EXAG (-0.67/-0.28). This validates a key property of effective cultural metrics: a faithful images must also penalize hallucination and exaggeration. This can be utilized by future work to validate adn constrcut stronger metrics. Additionally, ALIGN correlates positively with realism (0.51), indicating culturally aligned images also appear more photorealistic.

**Descriptor-based feedback enables targeted image editing.** Beyond evaluation, our descriptor-level feedback provides actionable guidance for improving generated images. Figure 8 demonstrates this capability across four activities. For each example, AHEaD identifies specific issues: hallucinated elements (e.g., elephants for "elephant ant man game," fur-leg leggings for Gumboot dance), exaggerated stereotypes (e.g., excessive batik clothing, Zulu attire with decorative beadwork), and aligned elements (e.g., standing in circle, players in close proximity). We use this feedback to create targeted editing prompts, instructing the model to remove hallucinated elements and reduce and diversify exaggeration while maintaining aligned elements (prompts are in 32). Edited images show substantial improvements, better matching real reference images by removing culturally incorrect elements and reducing stereotypical over-representation. This demonstrates that interpretable descriptor feedback enables iterative refinement toward more culturally faithful generation.

**Evaluation of LLM generated reference descriptors**. We validate reference descriptor quality through human annotation. Table 11 shows precision across 7 countries where annotators marked each GPT-4o-generated descriptor as correct or incorrect. Descriptors achieve high precision, aver-

| Method | Model | Flux1 | Qwen | SD3.5 | Avg. |
|---|---|---|---|---|---|
| Reference Descp ($\mathcal{D}^{ref}$) | EXAG (VQAscore) | -0.276 | -0.256 | -0.220 | -0.251 |
| Stereotype Cand. $S$ | EXAG (VQAscore) | 0.349 | **0.184** | 0.230 | **0.183** |
| MLLM-Desc. | EXAG (VQAscore) | -0.221 | -0.092 | 0.063 | -0.083 |
| Human | – | 0.231 | 0.097 | **0.482** | 0.270 |

Table 10: **Exaggeration per T2I (Spearman)**. Correlation across different text-to-image generators. Results are based on wcountries.

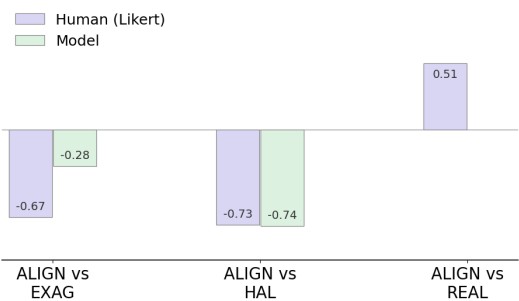

Figure 7: **ALIGN negatively correlates with HAL and EXAG.** Both human judgments (purple) and automatic metrics (green) show negative correlation between ALIGN (GT-FAITH for human) and HAL/EXAG (GT-HAL/GT-EXAG for human), confirming that effective faithfulness metrics must penalize hallucination and exaggeration. ALIGN positively correlates with realism.

aging 90.27% with minimum 85.21% (Iran), demonstrating reliable cultural accuracy across diverse regions.

Direct recall measurement is infeasible without ground-truth descriptor sets. We estimate recall through two measures: (1) annotators rated overall descriptor quality on a 5-point Likert scale (average: 4.5/5), and (2) annotators identified missing descriptors in free-text responses (only 26 of 378 reported any omissions). These results demonstrate high precision and comprehensive coverage. We further improve descriptor quality through our Proposer-Refiner approach (Section 3), as shown in Table 4.

**Raw human scores.** Table 13 summarizes human judgments across countries. Consistent with our automatic metrics, humans assign higher faithfulness scores (GT-FAITH) and lower exaggeration/hallucination scores (GT-EXAG/GT-HAL) for GN than GS countries. Interestingly, realism scores (GT-REALISM) are slightly higher for GS (3.31) than GN (3.15), though the difference is small. Standard deviations indicate substantial variation within countries, reflecting the subjective nature of cultural evaluation.

A.6    PROMPTS

In this section, we include prompts used in this project.

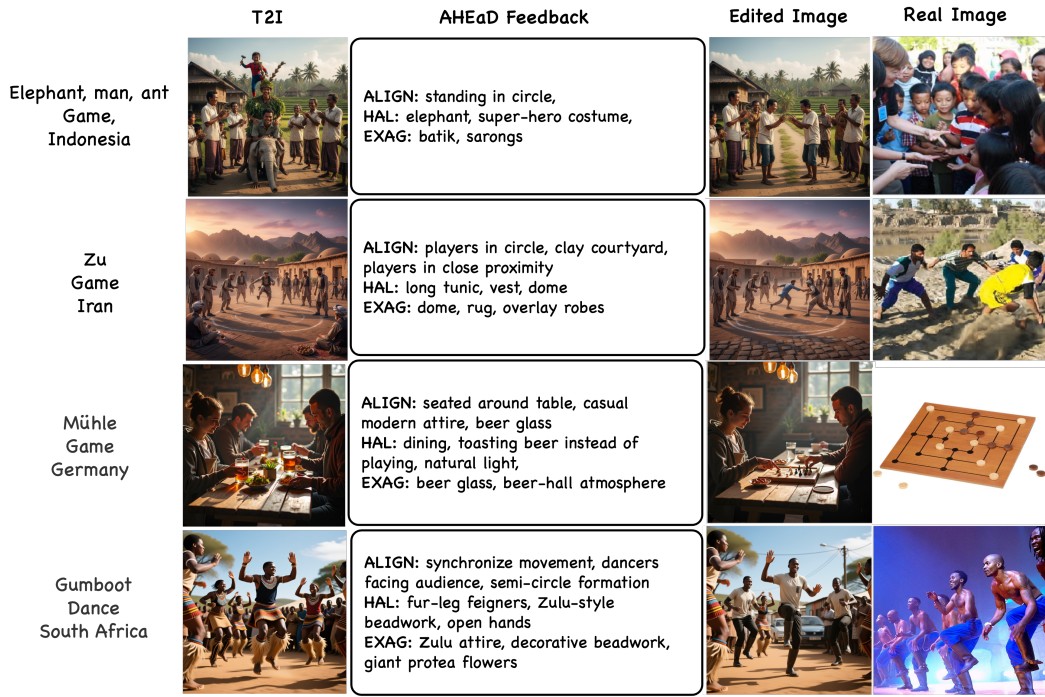

Figure 8: **Illustration of descriptor effectiveness in guiding image editing for improved generation.** (a) **Initial T2I-generated images** (top to bottom: Gemini 2.5 Flash Image, Gemini 2.5 Flash Image, FLUX, Qwen-Image). (b) **Generated feedback by AHEaD**: We use AHEaD feedback along with reference descriptors $\mathcal{D}^{ref}$ to create clear instruction prompts (prompt in Table.32) (c) **Edited images**: Nano-Banana is utilized to edit images according to instruction prompts generated in (b). (d) **Real images**.

| China | France | Iran | Nigeria | USA | India | Brazil | Avg. |
|-------|--------|------|---------|------|-------|--------|------|
| 89.80 | 90.54 | 85.21 | 91.62 | 91.44 | 91.61 | 91.68 | 90.27 |

Table 11: **LLM (GPT-4o) generated descriptors validation by humans.**

| Country | Activity | Top-3 Exaggerated Descriptors |
| --- | --- | --- |
| Brazil | Celebration | people in bikinis and swimwear; excessive Brazilian flags; favela backgrounds |
| Brazil | Eating | tropical rainforest backgrounds; excessive Brazilian flags; favela backgrounds |
| China | Celebration | pagoda-style roofs; traditional silk robes; red lanterns |
| China | Eating | red lanterns; oversized calligraphy scrolls; chopsticks in every scene |
| France | Celebration | café terraces; chateaux; Eiffel Tower |
| France | Eating | chandeliers; baguettes; croissants |
| Germany | Celebration | Gothic cathedrals; half-timbered houses; Alpine hats |
| Germany | Eating | bratwurst; cuckoo clocks; half-timbered houses |
| India | Celebration | exaggerated Bollywood posters; excessive gold jewelry; embellished saris |
| India | Eating | oversized diya lamps; exaggerated Bollywood posters; embellished saris |
| Indonesia | Celebration | batik clothing; exaggerated temple structures; Balinese gates |
| Indonesia | Eating | batik clothing; ubiquitous sarongs; large bamboo decorations |
| Iran | Celebration | giant domes; traditional robes; minarets in every scene |
| Iran | Eating | exaggerated calligraphy; giant domes; religious symbols |
| Italy | Celebration | Renaissance paintings; large Italian flags; exaggerated hand gestures |
| Italy | Eating | Renaissance paintings; excessive pasta dishes; wine bottles |
| Mexico | Celebration | Mexican flags everywhere; sombreros; papel picado |
| Mexico | Eating | papel picado; Frida Kahlo portraits; tequila bottles |
| Nepal | Celebration | giant pagoda roofs; Himalayan mountains; traditional attire |
| Nepal | Eating | prayer flags; Himalayan mountains; mandalas |
| Nigeria | Celebration | oversized bead necklaces; giant flags; market scenes |
| Nigeria | Eating | bead necklaces; gele headwraps; palm trees |
| Philippines | Celebration | Barong Tagalog worn by everyone; salakot hats; giant flags |
| Philippines | Eating | bamboo furniture; rice terraces; giant flags |
| South Africa | Celebration | oversized flags; large drums; Zulu beadwork |
| South Africa | Eating | oversized flags; township scenes; Zulu beadwork |
| Spain | Celebration | Gaudí-style architecture; oversized flags; red–yellow color schemes |
| Spain | Eating | red–yellow color schemes; Gaudí architecture; giant paella pans |
| Turkey | Celebration | minarets in every background; Ottoman architecture; oversized flags |
| Turkey | Eating | Ottoman architecture; oversized flags; Turkish carpets |
| USA | Celebration | oversized flags; suburban houses; cowboy hats |
| USA | Eating | fast food items; suburban houses; pickup trucks |

Table 12: **Example of Top-3 Exaggerated Descriptors**. Extracted exaggerated descriptors for each country–activity pair by EXAG metric. Descriptors reveal recurring cultural exaggerations associated with specific contexts and can be used to diagnose bias or guide model correction.

| Region | GT-FAITH | GT-EXAG | GT-HAL | GT-REALISM |
|---|---|---|---|---|
| FRANCE | 3.09 (0.73) | 2.63 (0.70) | 2.29 (0.74) | 3.25 (0.39) |
| BRAZIL | 3.61 (0.65) | 2.27 (0.47) | 1.79 (0.61) | 3.46 (0.20) |
| CHINA | 2.95 (0.84) | 2.66 (0.52) | 2.19 (0.52) | 3.17 (0.44) |
| INDIA | 3.27 (0.94) | 2.38 (0.73) | 1.97 (0.56) | 3.56 (0.65) |
| MEXICO | 2.91 (1.00) | 2.41 (0.86) | 2.38 (0.88) | 2.98 (0.65) |
| GERMANY | 3.25 (0.79) | 2.25 (0.59) | 2.04 (0.84) | 3.10 (0.52) |
| NIGERIA | 3.53 (0.73) | **1.97** (0.57) | 1.92 (0.52) | **3.75** (0.57) |
| TURKEY | 2.79 (1.22) | 2.40 (0.79) | 2.58 (1.18) | 3.33 (0.55) |
| USA | **3.96** (0.45) | 2.19 (0.47) | **1.59** (0.42) | 3.48 (0.52) |
| IRAN | 2.52 (0.76) | 3.03 (0.78) | 2.78 (0.53) | 3.15 (0.61) |
| SPAIN | 3.03 (1.10) | 2.25 (0.74) | 2.43 (0.68) | 2.95 (0.45) |
| **GS** | 3.04 (0.97) | 2.44 (0.74) | 2.28 (0.83) | **3.31** (0.59) |
| **GN** | **3.28** (0.89) | **2.31** (0.65) | **2.13** (0.75) | 3.15 (0.50) |

Table 13: **Human evaluation scores by country.** Mean (standard deviation) on 5-point Likert scale for faithfulness, hallucination, exaggeration, and realism. GN countries show higher faithfulness and lower hallucination/exaggeration than GS.

| Activity | Example Subactivities (across countries) |
|---|---|
| Eating | Home, Restaurant |
| Greeting | Namaste (India), Prostrating (Nigeria), Three-kiss (Iran), Cheek kiss (France) |
| Dancing | Samba (Brazil), Flamenco (Spain), Bharatanatyam (India), Dragon Dance (China) |
| Game | Kabaddi (India), Ayoayo (Nigeria), Pétanque (France), Baseball (USA), Mahjong (China) |
| Celebration | Nowruz (Iran), Carnival (Brazil), Bastille Day (France), Thanksgiving (USA), Chinese New Year (China) |
| Religious | Tazieh (Iran), Candomblé ceremony (Brazil), Catholic mass (Mexico), Temple aarti (India) |

Table 14: **A subset of examples of subactivities in CULTIVate.** Highlights distinctive cultural practices across countries.

---

**LLM Descriptor Generator — System Prompt**

**System**: You are an expert in cross-cultural visual representation. Your task is to generate precise visual descriptors capturing how a typical scene of a given activity appears in a specific country. Descriptors must cover both traditional and modern variations and represent common culturally accurate scenes.

**Rules**: 1. The output must strictly follow this JSON structure:
`"descriptors":["token":"...", "style":"traditional|modern|neutral"]`
2. Use culturally-aware terminology (e.g., samovar, sari) when appropriate; use broader cultural phrases when high specificity is unnecessary.
3. Focus only on the core activity scene (not before/after events).
4. Capture multiple common variations where they exist.
5. If the dimension has no representative descriptors, return an empty list.

---

Table 15: LLM descriptor generator – System Prompt

| LLM Descriptor Generator — Setting & Background |
| --- |
| **Goal**: Describe the environment — the physical location, architecture, and design elements that define the atmosphere of the scene. 
 **Guidelines**: **INCLUDE**: 
 – Location and architectural style (indoors/outdoors; temple interior, city street) 
 – Art and design (calligraphy, geometric tiles, minimalist décor) 
 – Major furnishings (communal tables, floor cushions, rugs) 

 **EXCLUDE**: people, clothing, handheld objects, specific actions. 

 **Generate up to** {max_items} **descriptors for**: 
 {concept} |

Table 16: LLM descriptor generator – Setting & Background

| LLM Descriptor Generator — Objects |
| --- |
| **Goal**: Identify the core objects central to the activity. 
 **Guidelines**: 
 - Ensure descriptors accurately represent objects common in the activity scene within the given country. 
 - **INCLUDE**: essential tools, vessels, foods (samovar, board game, hot pot). 
 - Use visually descriptive categories (e.g., "bowls of noodle soup") instead of abstract labels ("Chinese food"). 
 **EXCLUDE**: people, animals, clothing, architecture, actions, background décor. 
 **Generate up to** {max_items} **descriptors for**: 
 {concept} |

Table 17: LLM descriptor generator – Objects

| LLM Descriptor Generator — Attire |
| --- |
| **Goal**: Describe typical clothing, accessories, and appearance features. 
 **Guidelines**: - Use specific garment names only when culturally essential (e.g., sari). 
 - Otherwise, use broader cultural categories (e.g., traditional West African attire). 
 - Include both traditional and modern clothing variations unless the concept is strictly historical. 
 **INCLUDE**: garments, headwear, accessories, ceremonial markings, uniforms. 
 **EXCLUDE**: tools, furniture, actions, gestures. 

 **Generate up to** {max_items} **descriptors for**: {concept} |

Table 18: LLM descriptor generator – Attire

| LLM Descriptor Generator — Interaction & Gesture |
| --- |
| **Goal**: Capture actions, gestures, and social dynamics central to the activity. 
 **Guidelines**: **INCLUDE**: 
 – Key person–object actions (pouring tea from samovar) 
 – Social gestures (sharing food, group dancing) 
 – Culturally typical postures and formations (kneeling rows) 
 **EXCLUDE**: static object descriptions, clothing, setting details. Focus on actions and interactions. 

 **Generate up to** {max_items} **descriptors for**: {concept} |

Table 19: LLM descriptor generator – Interaction & Gesture

**LLM Descriptor Generator — Spatial Arrangement**

**Goal**: Describe layout and spatial organization of people and objects.
**Guidelines**: **INCLUDE**:
- Positioning of people relative to key objects or surfaces
- Culturally meaningful configurations (eating at a table vs. around a sofreh)
- Ensure descriptors cover common variations in the activity across the country.

**EXCLUDE**: clothing details, object descriptions, actions.
**Generate up to** {`max_items`} **descriptors for**: {`concept`}

Table 20: LLM descriptor generator – Spatial Arrangement

**LLM Refiner Prompt**

**System**: You refine candidate visual descriptors for evaluating the cultural alignment of AI-generated images representing a specific concept or activity in a given country. Your job is to select, clean, and filter descriptors based on cultural accuracy and relevance.

**Task**: Select and refine descriptors according to the concept, country, and descriptor dimension.

**Dimensions**:
- Setting — venues, architecture, décor
- Objects — central objects in the activity
- Attire — clothing, accessories, headwear
- Interaction — gestures, postures, social relations
- Spatial Layout — positioning patterns

**Rules**: 1. Keep only culturally accurate descriptors.
2. Create a diverse set covering typical variations.
3. Do not invent new descriptors.
4. Merge duplicates or overly specific items.
5. Remove unrelated descriptors.
6. Keep phrases concise (1–4 words).
7. Descriptors must match the assigned dimension.
8. Output up to {`max_items`} descriptors.
9. If none are valid, return an empty list.

**Output Format**: `["token":"item","style":"traditional|modern|neutral"]`

**Input**: Concept: {`prompt`} in {`country`} Dimension: {`dimension`} Candidate Descriptors: {`candidate_descriptors`}

Table 21: LLM Refiner Prompt

**MLLM Descriptor Exctractor (System Prompt)**

As an expert on cross-cultural visual representation, your task is to generate precise visual descriptors to evaluate the cultural alignment and accuracy of AI-generated images.
**Goal**: Capture visual elements of a typical scene of an activity in a specific country, covering both traditional and modern variations.
**Rules**: 1. Output strictly in JSON: `"descriptors":["token":"...", "style":"traditional|modern|neutral"]`
2. Use culturally-aware terms (e.g., samovar, sari) when precise, or broader cultural terms when sufficient.
3. Focus on the core activity scene—not before or after actions.
4. Capture common variations (e.g., eating at a table vs. sitting on the floor).
5. If nothing distinctive exists, return an empty list.

Table 22: System Prompt for descriptor generation.

**MLLM Descriptor Extractor (Setting & Background Prompt)**

**Goal**: Describe the environment (location, architecture, design, furnishings).
**INCLUDE**:
- Indoors/outdoors (temple interior, busy street, simple home)
- Art & design (calligraphy, tiles, minimalist decor)
- Major furnishings (floor cushions, rugs, communal tables)

**EXCLUDE**: clothing, handheld objects, actions.

Table 23: MLLM descriptor detector (Setting & Background).

**Objects Prompt**

**Goal**: Identify key objects, tools, foods, vessels central to the activity.

**INCLUDE**: essential items (samovar, board game, noodle bowls, shared hot pot).

**EXCLUDE**: animals, clothing, architecture, actions, background décor.

Table 24: Prompt: Objects.

**Attire Prompt**

**Goal**: Describe typical clothing, accessories, and appearance.

**Rules**:
- Use specific garment names only when culturally essential (e.g., sari).
- Otherwise, use broader cultural categories.
- Always include both traditional and modern possibilities.

**INCLUDE**: garments, headwear, accessories, ceremonial markings, uniforms.
**EXCLUDE**: tools, furniture, actions, gestures.

Table 25: MLLM descriptor detector (Attire).

**Interaction & Gesture Prompt**

**Goal**: Capture actions, gestures, and social dynamics.

**INCLUDE**:
- Person and/or object actions (pouring tea from a samovar)
- Social gestures (sharing food, group dancing)
- Group formations (kneeling rows, circle formations)

**EXCLUDE**: static objects, clothing, setting.

Table 26: Prompt: Interaction & Gesture.

**MLLM Descriptor Detector (Spatial Arrangement)**

**Goal**: Describe the physical layout and positioning of key objects.

**INCLUDE**:
- Relative positions (sitting around sofreh, standing in line)
- Culturally significant layouts (table seating vs. floor seating)

**EXCLUDE**: clothing, object details, gestures.

Table 27: MLLM descriptor extractor (Spatial Arrangement).

---

**Prompt**

---

**System**: You are a helpful assistant that identifies culture-specific visual elements that a text-to-image model may exaggerate when depicting a given activity in a given country. These elements should represent unrealistic overuse, oversized forms, or out-of-place insertions of cultural cues. Return only concise visual stereotypes based on observable features.

**Task**: Generate up to {{num_elements}} candidate exaggerated elements for the activity and country below. Elements must be concise noun phrases describing culture-specific items (e.g., "Persian rugs", "giant flags"). Focus on attire, objects, symbols, decorations, architecture, and interaction styles. Avoid generic or non–culture-specific items (e.g., "trees", "crowds"). Do not describe scenarios or contexts—only list the exaggerated element itself.

ACTIVITY: {{concept}}
COUNTRY: {{country}}

**User**: Provide the final list as a single line of noun phrases separated by the "—" character. Do not include numbering, explanations, or extra text.

Example format:
```
Persian rugs | oversize lanterns | giant religious symbols
| people in traditional robes
```

---

Table 28: EXAG Candidate Generation Prompt

---

**ALIGN Likert Scoring Prompt**

---

**System**: You are a helpful assistant scoring how well an image depicts an activity as it would typically appear in the specified country. Provide a rating from 1 to 5 using the guidelines below. Only output the score in the required format.

**Scoring Scale**:
1 = Not well at all (completely wrong or unrelated)
2 = Slightly relevant
3 = Somewhat accurate
4 = Mostly accurate
5 = Exactly as expected for this country

**Task**:
PROMPT: {{prompt}}
COUNTRY: {{country}}

**User**: Respond with ONLY the score in the exact format below:
score is $< num >$

Do not include any other text, explanation, or formatting.

---

Table 29: ALIGN MLLM-as-a-Judge Prompt

**HAL MLLM-as-a-Judge Prompt**

**System**: You are a helpful assistant scoring whether an image contains incorrect elements with respect to the mentioned activity and country. Rate on a scale from 1 to 5 using the rules below. Only output the score in the required format.

**Scoring Scale**:

1 = Not at all. All elements seem appropriate.

2 = Slightly. One minor element seems foreign.

3 = Somewhat. A few elements feel unusual.

4 = Mostly. Many elements appear incorrect or from another culture.

5 = Strongly. The image is completely wrong, most elements are wrong, the activity is wrong, or the scene belongs to another country.

**Task**:
PROMPT: {{prompt}}
COUNTRY: {{country}}

**User**: Respond with ONLY the score in the exact format below:
score is $< num >$

Do not include any other text, explanation, or formatting.

Table 30: HAL MLLM-as-a-Judge prompt

**EXAG MLLM-as-a-Judge Prompt**

**System**: You are a helpful assistant scoring how exaggerated an image is in terms of cultural elements (e.g., attire, objects, background, art/design). Rate on a scale from 1 to 5 using the rules below. Only output the score in the required format.

**Scoring Scale**:

1 = Appropriate and balanced.

2 = Slightly decorative but within cultural norms.

3 = Several features feel idealized or overly emphasized.

4 = Many elements feel exaggerated and unrealistic.

5 = Strongly stereotypical, misleading, or culturally irrelevant.

**Task**:
PROMPT: {{prompt}}
COUNTRY: {{country}}

**User**: Respond with ONLY the score in the exact format below:
score is $< num >$

Do not include any other text, explanation, or formatting.

Table 31: EXAG MLLM-as-a-Judge Prompt

---

**Image Editing Instruction Prompt**

---

**Task**: Edit the image to correctly show {{`activity`}} in {{`country`}} by following the instructions below.

**Remove**:

- All hallucinated elements from the HAL list

- All exaggerated elements from the EXAG list

- Any objects, clothing, poses, or background features that belong to the wrong culture, historical period, or activity

**Add/Preserve**:

- ALIGN list that must remain present

- Add correct interaction from REF DESCP list if mentioned in HAL/EXAG

- ADD 1-3 different types of attire from REF DESCP if mentioned in HAL/EXAG

- ADD 1 correct background from REF DESCP if mentioned in HAL/EXAG

**Input**:

ACTIVITY: {{`activity`}}

COUNTRY: {{`country`}}

HAL DESCRIPTORS: {{`HAL`}}

EXAG DESCRIPTORS: {{`EXAG`}}

ALIGN DESCRIPTORS: {{`ALIGN`}}

REFERENCE DESCRIPTORS: {{`REF DESCP`}}

---

Table 32: Image editing instruction prompt template. AHEaD feedback (HAL, EXAG, ALIGN) combined with reference descriptors $\mathcal{D}^{ref}$ guides image editing to improve cultural accuracy.

