# OpenReview forum: "Culture in Action: Evaluating Text-to-Image Models through Social Activities"
_ICLR.cc/2026/Conference — ICLR 2026 Poster_

### Official Review · Reviewer_yiqA · 2025-10-26

**Soundness:** 3
**Presentation:** 3
**Contribution:** 2
**Rating:** 4
**Confidence:** 4

**Summary:**

The paper builds a benchmark and evaluation framework to measure cultural faithfulness of T2I models when prompted to generate social activities. It also introduces an interpretable evaluation metric capturing 4 diagnostic aspects which correlates better with human judgements compared to previous metrics or using a model as a judge. The proposed pipeline uses a proposer-refiner approach for generating and verifying textual descriptions. The experiments show cultural disparities between Global North and Global South depictions and highlight failure modes.

**Strengths:**

1. The paper is well-written and clear. The main contribution of the paper lies in the metric, which is explainable and actionable through the decomposition of descriptor dimensions. The metric definitions (ALIGN, HAL, EXAG, DDIV/SDIV) are also well-defined and motivated.
2. The paper includes many ablations (e.g., τ threshold, proposer vs proposer+refiner) and per-dimension analyses.
3. There is strong evidence of the metric utility through a higher Spearman correlation with human GT labels than common ITA baselines and the model as a judge.
4. The use case of the metric is also clear, since it enables interpretable comparison across different aspects of various T2I models.

**Weaknesses:**

1. The empirical findings are incremental, several concurrent works (though reported by the authors) report similar Global North vs Global South gaps and also collect prompts and images for social activities.
2. The same MLLM (InternVL) is used to extract image descriptors in the pipeline (L727) and also appears in baseline comparisons (InternVL as a judge). This creates a potential dependency/leakage where the extraction backbone advantages or disadvantages affect both measurement and baselines.
3. Human evaluation was conducted for 7 countries × 9 activities × 3 models plus some real images (216 forms total). This is a small subset relative to the full 576 prompts / 16 countries. The paper reports Spearman correlation numbers and "human-human agreement is moderate (L424)" but lacks thorough reporting of annotator reliability (Krippendorff’s α / Cohen’s κ), per-country inter-annotator agreement, and statistical significance of the reported metric improvements.
4. EXAG depends on LLM-produced “exaggeration candidates” and on web images as references. It is unclear how candidates are generated, how stereotypical or noisy web images affect EXAG, and how to distinguish a valid cultural variant from an exaggeration.
5. The GN/GS split (listed countries) is reasonable for summarizing trends, but grouping heterogeneous countries into GS may obscure important nuances. For instance, China often performs well despite being placed in GS.

**Questions:**

1. It will be intuitive to see some concrete examples of descriptors that the refiner removed. For descriptors that annotators marked irrelevant (from Table 9), what categories of errors occur (mislabeling, overly generic, stereotype)?
2. To address W2, it will be helpful to run the descriptor-extraction with 1–2 additional independent MLLMs (e.g., QwenVL or another open model) and report sensitivity of ALIGN/HAL/EXAG to the extractor choice. If results are stable, it will strengthen robustness claims, but if not, discuss limitations.
3. How do the results look at per-country level or per-activity level? Which countries drive the gap? Why do particular countries (in GS bucket) perform better/worse?
4. Is the 7% improvement with human judgment statistically significant? Report confidence intervals / p-values for the correlation improvements.

---

> ### Author Response · Authors · 2025-11-26
> **response (part 1/2)**
>
> We thank Reviewer yiqA for their thoughtful and positive assessment. We appreciate that they found the paper **well-written** and clear, the **metric explainable**, **well-motivated**, **actionable**, with a **clear use case and strong interpretability**. We are also glad they highlighted our **comprehensive ablations** and the **strong evidence of metric effectiveness**, including the high correlation with human judgments.
> We respond to each comment inline:
>
>
> **The empirical findings are incremental ... concurrent works (reported by authors) report similar Global North vs Global South gaps and also collect prompts...for social activities.**
>
> Thanks, we would like to clarify that our contributions extend beyond GN/GS disparities. We demonstrate that alignment alone is inadequate for cultural faithfulness and establish properties for effective metrics: negative correlation with exaggeration and hallucination (Fig. 5). We provide the first automatic metrics for cultural faithfulness (HAL, EXAG, ALIGN, FAITH)-- prior work relied solely on human evaluation. Additionally, we offer interpretable descriptor-level feedback identifying which elements are aligned, hallucinated, or exaggerated, enabling targeted model improvements. These contributions differ fundamentally from concurrent work focused on documenting disparities through expensive-to-collect human evaluation.
>
>
> **The same MLLM (InternVL) is used to extract image descriptors in the pipeline (L727) and also appears in baseline comparisons (InternVL as a judge). This creates a potential dependency/leakage where the extraction backbone advantages or disadvantages affect both measurement and baselines.**
>
> We respectfully clarify there is no leakage.**InternVL-as-judge and our metric are completely independent**: InternVL-as-judge directly evaluates images for cultural faithfulness by answering the same questions we ask human annotators (L419) and **never sees descriptors**.  We compare ours (with InternVL) with InternVL-as-a-Judge to evaluate whether our proposed descriptor-descriptor is more effective than just using the MLLM directly for cultural faithfulness? Results show FAITH (InternVL) is significantly (+0.27) better than MLLM baseline (Table 2)
>
>
> **Human evaluation ...a small subset relative to the full ... ...reporting of annotator reliability (Krippendorff’s α / Cohen’s κ), per-country inter-annotator agreement**
>
> Thank you for insightful feedback. We have expanded our human evaluation to 11 countries with 2 prompts per activity, nearly **doubling annotations**(see [G1]). The original study covered representative countries (all regions, both GS/GN) and all 9 activity categories.
>
> Inter-rater reliability (We revised A.4): We measured Krippendorff's α per country (range: 0.15-0.62) and Cohen's κ between two annotator groups (mean: 0.50). These are **comparable to prior cross-cultural work**: CulturalFrames [2] reports Krippendorff range of 0.24-0.42 and CUBE [3] reports 0.09-0.58 (numbers borrowed from [2]). We observe comparable agreement to related works. Variation reflects that cultural faithfulness judgments are highly subjective, particularly across different countries and cultural contexts.
>
>
> **EXAG ... It is unclear how candidates are generated, how ... noisy web images affect EXAG, and how to distinguish a valid cultural variant from an exaggeration.**
>
> Thanks. Inspired by [1], we use LLMs to generate exaggeration candidates (added prompts in Appendix). We validate this approach in Table 11 in revised version (Table 10 in original version), comparing 3 strategies: (1) LLM reference descriptors, (2) real image descriptors, and (3) exaggeration candidates. Results show exaggeration candidates achieve significantly better human correlation (eg., 0.35 vs. -0.27 of LLM ref. descriptors on FLUX ), confirming they capture stereotypical over-representation.
>
> *Distinguishing variants from exaggeration:* **Exaggerated descriptors are **valid** but over-represented **(e.g., excessive traditional wall designs). We measure this by comparing element frequency in generated vs. real images. Invalid descriptors are captured by HAL (Sec. 3).
>
> We acknowledge real image quality affects EXAG (discussion in A.3). This is the first work quantifying exaggeration for cultural evaluation; we also explored MLLM-as-judge as an alternative not dependent on real image quality. We hope this work inspires development of stronger metrics.
>
>
> **The GN/GS split is reasonable ... but ... may obscure important nuances. For instance, China often performs well despite being placed in GS.**
>
> We agree that including per country results is beneficial. We clarify that Global South/Global North grouping is based on United Nation Classification.
> We also show per-country results in Fig.3a  (on ALIGN and HAL score) and Fig.4 across different dimensions which in fact show that China data is more representative. For instance Qwen-Image even supports the Chinese language.

---

> ### Author Response · Authors · 2025-11-26
> **response (part 2/2)**
>
> **intuitive to see concrete examples of descriptors that the refiner removed …. what categories of errors occur (mislabeling, overly generic, stereotype)?**
>
> Thank you. We provided an example of descriptors being removed by LLM reference refiner in Fig. 1 and will include more in the Appendix.
>
> Error categories from human evaluation (Table 10 in Appendix): The 10% marked irrelevant includes mislabeled/culturally incorrect concepts. Note that generic or stereotypical descriptors are not removed as they are not incorrect. For example, "muslim attire" is often exaggerated in Iran but not incorrect (people can be Muslim). Other examples of problematic descriptors (e.g. hallucinated or exaggerated) appear in the supplementary file. For example, see the top-3 automatically extracted exaggerated descriptors, with their respective country/activity (shown for only two of our activities), in response (part 2/3) to reviewer b2M6; subset of examples shown here. This motivated our metrics and shows our intuition that alignment alone is incorrect. Hence we purpose HAL and EXAG to capture incorrect and exaggerated elements in the image.
>
> Brazil/celebration: people in bikinis and swimwear, excessive Brazilian flags, favela backgrounds
>
> Brazil/eating: tropical rainforest backgrounds, excessive Brazilian flags, favela backgrounds
>
> China/celebration: Pagoda-style roofs, Traditional silk robes, Red lanterns
>
> China/eating: Red lanterns, Oversized calligraphy scrolls, Chopsticks in every scene
>
> France/celebration: café terraces, chateaux, Eiffel Tower
>
> France/eating: chandeliers, baguettes, croissants
>
> Nigeria/celebration: oversized bead necklaces, giant Nigerian flags, exaggerated market scenes
>
> Nigeria/eating: oversized bead necklaces, Giant gele headwraps, ubiquitous palm trees
>
> Philippines/celebration: Barong Tagalog worn by everyone, everyone wearing salakot hats, giant Philippine flags
>
> Philippines/eating: excessive bamboo furniture, overabundance of rice terraces, giant Philippine flags
>
> South Africa/celebration: oversized South African flags, large African drums, Zulu beadwork
>
> South Africa/eating: oversized South African flags, exaggerated township scenes, Zulu beadwork
>
>
> **To address W2, it will be helpful to run the descriptor-extraction with 1–2 additional independent MLLMs (e.g., QwenVL or another open model) and report sensitivity of ALIGN/HAL/EXAG to the extractor choice. If results are stable, it will strengthen robustness claims, but if not, discuss limitations.**
>
>  In Table 2 (revised) we compare our approach with two MLLMs (InternVL3 and QwenVL2.5) and corresponding MLLM-as-a-Judge with the same backbone. Results show AHEaD with InternVL3 backbone achieves 0.47 correlation (vs 0.20 for InternVL3-as-judge), and AHEaD with QwenVL2.5 achieves 0.42 (vs 0.10 for QwenVL2.5-as-judge on GT-Faith (overall). This show that our metric doesn't rely on cultural understanding of the MLLMs. We observe similar trend on GT-HAL in Table 6. Also, this analysis doesn't apply to descriptor-based EXAG as it doesn't use an MLLM.
>
>
> **How do the results look at per-country level or per-activity level? Which countries drive the gap? Why do particular countries (in GS bucket) perform better/worse?**
>
> Thank you for this suggestion. We'd already provided analysis in Fig. 3a and Fig. 3b provides per-country (ALIGN/HAL) and per-activity (ALIGN) analysis. Top performers are China, USA, Germany, Italy, and Spain; bottom performers are Indonesia, Nigeria, India, South Africa, and Iran. Except for China, models consistently perform better in GN vs GS countries. We conjecture China's performance stems from higher web data availability, particularly for Qwen-Image which even supports Chinese (China is GS per United Nation classification). Per-activity results (Fig. 3b) show models struggle with culturally-specific activities (celebrations, local games, dances) but perform well on generic activities (concerts, modern weddings). We will expand this analysis in Sections 5.3-5.4 and provide detailed breakdowns in the appendix.
>
>
> **Report confidence intervals / p-values for the correlation improvements.**
>
> Thank you for this important question. We've updated results with expanded human evaluation (2x more prompts). In Table 3, our ALIGN significant p-values <0.0001 (e.g.1.68e-10 (GS), 2.95e-07 (GN), 3.98e-17 (Overall)). We also clarify that according to the updated table, ALIGN alone achieves up-to 20% higher correlation with human faithfulness in Table 3.  We will update the paper reflecting new numbers and statistical results (p-values and confidence intervals) in the next version.
>
>
> [1] OpenBias: Open-set Bias Detection in Text-to-Image Generative Models, CVPR 2024
>
> [2] CulturalFrames: Assessing Cultural Expectation Alignment in Text-to-Image Models and Evaluation Metrics
>
> [3] Beyond Aesthetics: Cultural Competence in Text-to-Image Models

---

### Official Review · Reviewer_b2M6 · 2025-10-30

**Soundness:** 3
**Presentation:** 2
**Contribution:** 2
**Rating:** 6
**Confidence:** 5

**Summary:**

This work introduces CULTIVate, a new cultural activity focused benchmark for cultural faithfulness of text-to-image models spanning 16 countries and 576 prompts. They use cultural descriptors generated by LLMs to create AHEaD metrics to measure cultural alignment, hallucination, exaggerated elements (stereotypes), and descriptor diversity. They find T2I models are more aligned to the cultures of countries from the Global North than the Global South, and show strong Spearman rank correlation of their proposed AHEaD metrics to human judgments via a user study.

**Strengths:**

* This works makes contributions towards cultural benchmarking of text-to-image models, which is a relatively underexplored and important area of research
* They provide a new benchmark that is focused on cultural activities in contrast to prior work that focuses more on cultural objects (e.g. landmarks). The benchmark size (576 prompts), categorical coverage (9 super categories of activities) and regional coverage (16 countries) is of similar order to prior work [1- 3, 5].
* The proposed AHeAD metrics are based on interpretable cultural descriptors (Fig 4) and correlate well with human judgments when compared to baseline metrics of image-text alignment and multimodal LLM judges (Tab. 2)

Note: see references in "Weaknesses"

**Weaknesses:**

## Major Weaknesses:

* I disagree with a central claim of the authors, i.e. that their benchmark "... capture the contextual complexity **missing from object-centric benchmarks**" (L74). Prior work has already evaluated on contextually-specific cultural activities, e.g. celebrations / festivals / performance arts [1- 5], weddings [4], religious activities [1], sports [1, 2, 6], cooking [6]. The authors claim that their activity-focused benchmark highlights "**new failure patterns**" such as "hallucinated elements" or "heavily exaggerated scenes" (L76 - 78). This behavior has also been established in prior work that discusses culture-specific stereotyping in detail [1, 2, 7]. They also claim that prior work [1, 3] rely on VLM metrics that inherit similar cultural biases, however these works also use visual perceptual similarity and diversity based metrics that do not use VLMs. In fact Rege et al. stresses on this scorer bias ('generative entanglement'), which is not mentioned in this work.

* The authors stress that prior metrics using VLMs will inherit their biases (L85), but use multimodal LLMs (GPT-4o and Gemini 2.5 Flash) to generate culturally-specific visual descriptors (L206 - L215). Since these MLLMs are also trained on internet data, will they not also inherit similar biases as VLMs? Since the authors have highlighted VLM biases, they should also discuss if MLLMs are similarly biased, as this directly affects their AHeAD metrics. If they are not similarly biased, the authors should point to empirical evidence showing so explicitly in the paper.

* For EXAG, the authors use an LLM to generate *"exaggeration candidates"* (L245). How do we know that these are culturally accurate? Are these candidates rated by humans who identify with these cultures?

* When comparing against image-text alignment baselines (Tab 2), what prompts are used? What configuration of CuRe I-T alignment scorer [3] was used? What prompts are used for the MLLMs (InternVL and QwenVL?) These details are very important to establish a fair comparison to baselines.

## Minor Weaknesses:

* The paper is generally written well, but has several vague descriptions that make the contributions hard to follow:
    * L117: *"Our AHEaD metrics achieve 7% better correlation with human judgment"* - better than what, and what are the humans judging?
    * L118: *"The best agreement is achieved ... "* - agreement between what?
    * L212: *"...maximizing recall of cultural elements"* - recall with respect to what? In my understanding, there is no ground truth descriptor set to which the LLM predictions can be compared to?

* In their user study, the authors mention that they "conduct direct discussions with annotators when facing inconsistent scoring and explanations"- what does this mean? Do the authors instruct these annotators on how to "correctly" score or write explanations?

* Fig. 3a is quite hard to visualize; the legend colors do not match, and one metric is better if higher and the other if lower. Consider using a side-by-side bar chart for this.

### References
---
[1] Khanuja et al., An image speaks a thousand words, but can everyone listen? on image transcreation for cultural relevance. *In EMNLP, 2024.*

[2] Zhang et al., Partiality and misconception: Investigating cultural representativeness in text-to-image models. *In CHI, 2024.*

[3] Rege et al., CuRe: Cultural Gaps in the Long Tail of Text-to-Image Systems. *In ICCV, 2025*.

[4] Basu et al., Inspecting the geographical representativeness of images from text-to-image models. *In ICCV, 2023*.

[5] Kannen et al., Beyond aesthetics: Cultural competence in text-to-image models. *In NeurIPS D&B Track, 2024*.

[6] Romero et al., Cvqa: Culturally-diverse multilingual visual question answering benchmark. *In NeurIPS D&B Track, 2024*.

[7] Jha et al., Visage: A global-scale analysis of visual stereotypes in text-to-image generation. *In ACL, 2024.*

**Questions:**

* How do the authors use the "free text rationales" from the user study (L354)?
* How many annotators provide ratings for each form (L341)? This is important to know how many people agree / disagree about ratings.
* It would be nice for L32 - 36 in the introduction to have citations to prior studies showing the effort and importance of cultural adaptation
* What is used to compute sentence embedding similarity in the ALIGN metric (L235)? Is this metric also culturally biased?
* The prompts in the Appendix (Tab 12 - 17, 26) are cut off and unreadable in full

---

> ### Author Response · Authors · 2025-11-26
> **response (part 1/3)**
>
> We thank Reviewer b2M6 for their constructive and positive review. Specifically, we’re glad they found our **benchmark new vs. object-centric prior work**, contributing to the **underexplored area** of cultural benchmarking in T2I models, with **appropriate scale and regional coverage**. We are also pleased they found our AHEaD **metrics interpretable and well-correlated with human judgments** and our **paper well-written**.
> We respond to each comment inline:
>
> **I disagree with a central claim of the authors, i.e. that their benchmark "... capture the contextual complexity missing from object-centric benchmarks" (L74)**
>
> While the cited works contain **very minimal** activity-related prompts, they are mainly object-centric in benchmark, analysis, and evaluations, and we focus on aspects like activities, social norms, interaction among people, not captured in these works. Bellow we provide per paper differences (while these are mainly are already acknowledged in Sec. 2, we'll update to have a more comprehensive discussion):
>
> [R1] studies transcreation (adapting one concept from culture to another; concepts are generic or related to only source culture), is mainly object-centric, and different from us doesn't quantitatively evaluate interaction, social norms, etc. [R2] includes a few activity categories but their focus is only on "cultural objects" (i.e., architecture, food, apparel; Table 6 of their paper) through Cultural Objects benchmark (UCOGC). [R3] focuses on mainly object-oriented categories like arch.,food, fashion from Wikipedia hierarchies (only includes celebration in some countries), but we focus on activities balanced across countries, and focus on interactions, social norms, multi-person activities. In Table 2, we showed that [R3] achieves 0.1 rank correlation with human judgement of faithfulness on our benchmark. In [R4] only 2/10 nouns are related but they are not culturally specific. Eg., they study "festival in Germany" but we study "Oktoberfest in Germany", more rigorously evaluating cultural competence. [R5] includes 3 types of concepts : Cuisine, landmarks, and arts (see Table 13; page 29) across only 8 countries (vs 16 ours). Only 1 subgroup of art (i.e. performance art) is related, and still lacks discussion of unique challenges in activities (e.g., interaction such as bowing when greeting in China). [R6] is a VQA benchmark, doesn't study bias in generative models (cross-cultural studies are more extensive in understanding domain; concurrent; L153)
>
> Different from these works, our benchmark parses Wikipedia and CulturalAtlas to benchmark, culturally specific multi-person activities and social norms where roles, gestures, interaction structure, and etiquette norms are evaluated through a balanced set of prompts across several categories. In addition, our framework introduces descriptor-based evaluation, provides first insights on properties of strong cultural relevance metric while prior works rely on human annotations, and proposes new metrics to capture different culturally-specific dimension (interaction, spatial, objects, scene/setting, attire) unique to compositional scenes such as activities. Our descriptor-based approach is interpretable and can be used to directly improve results, e.g. through image editing.
>
>
> **"hallucinated elements" or "heavily exaggerated scenes" …  has also been established in prior work that discusses culture-specific stereotyping in detail [1, 2, 7].**
>
> Thank you for this comment. We clarify our unique contribution: we are the first to propose automatic metrics quantifying hallucination and exaggeration. Critically, stereotypes and exaggeration are distinct concepts. Stereotypes are biased associations over distributions (e.g., most Middle Eastern men depicted with beards), while exaggeration is over-representation within a single image (e.g., excessive rugs, everyone in traditional clothing). Ten images of regular bearded men is stereotypical but each image may or may not be exaggerated; one image with excessive cultural elements (e.g. rugs) is exaggerated regardless of distribution. Table 2 and Fig. 5 show FAITH(ALIGN+HAL+EXAG) achieves 0.47 rank correlation vs ~0 for ITA baselines, demonstrating these metrics are essential for cultural evaluation.
>
> The cited papers analyze different phenomena and do not quantify our specific failure patterns. [2] focuses on object misrepresentation, not scene-level hallucination or exaggeration. [1] reports qualitative stereotyping in image editing but provides no quantitative metrics for hallucination or exaggeration. [7] measures identity stereotypes in portraits (e.g., gender, occupation), we study a different task (activities), show distinct limitations. Moreover, this the first work providing insights and proposing metrics for cultural faithfulness (FAITH)

---

> ### Author Response · Authors · 2025-11-26
> **response (part 2/3)**
>
> We note the top-3 automatically extracted exaggerated descriptors, with their respective country/activity (shown for only two of our activities). Note these are extracted automatically, which makes our work scalable, and are interpretable and useful for improving models, e.g. through image editing.
>
> Brazil/celebration: people in bikinis and swimwear, excessive Brazilian flags, favela backgrounds
>
> Brazil/eating: tropical rainforest backgrounds, excessive Brazilian flags, favela backgrounds
>
> China/celebration: Pagoda-style roofs, Traditional silk robes, Red lanterns
>
> China/eating: Red lanterns, Oversized calligraphy scrolls, Chopsticks in every scene
>
> France/celebration: café terraces, chateaux, Eiffel Tower
>
> France/eating: chandeliers, baguettes, croissants
>
> Germany/celebration: Gothic cathedrals, Half-timbered houses, Alpine hats
>
> Germany/eating: Bratwurst, Cuckoo clocks, Half-timbered houses
>
> India/celebration: exaggerated Bollywood posters, excessive use of gold jewelry, saris with excessive embellishments
>
> India/eating: oversized diya lamps, exaggerated Bollywood posters, saris with excessive embellishments
>
> Indonesia/celebration: batik clothing, exaggerated temple structures, traditional Balinese gates
>
> Indonesia/eating: batik clothing, ubiquitous sarongs, large-scale bamboo decorations
>
> Iran/celebration: giant domes, people in traditional robes, minarets in every scene
>
> Iran/eating: exaggerated calligraphy, giant domes, giant religious symbols
>
> Italy/celebration: Renaissance paintings, large Italian flags, exaggerated hand gestures
>
> Italy/eating: Renaissance paintings, excessive pasta dishes, wine bottles everywhere
>
> Mexico/celebration: Mexican flags everywhere, Sombreros, colorful papel picado
>
> Mexico/eating: colorful papel picado, Frida Kahlo portraits, tequila bottles
>
> Nepal/celebration: giant pagoda roofs, Himalayan mountain backdrop, people in traditional Nepali attire
>
> Nepal/eating: Prayer flags, Himalayan mountain backdrop, large mandalas
>
> Nigeria/celebration: oversized bead necklaces, giant Nigerian flags, exaggerated market scenes
>
> Nigeria/eating: oversized bead necklaces, Giant gele headwraps, ubiquitous palm trees
>
> Philippines/celebration: Barong Tagalog worn by everyone, everyone wearing salakot hats, giant Philippine flags
>
> Philippines/eating: excessive bamboo furniture, overabundance of rice terraces, giant Philippine flags
>
> South Africa/celebration: oversized South African flags, large African drums, Zulu beadwork
>
> South Africa/eating: oversized South African flags, exaggerated township scenes, Zulu beadwork
>
> Spain/celebration: Gaudí-style architecture, oversized Spanish flags, red and yellow color schemes
>
> Spain/eating: red and yellow color schemes, Gaudí-style architecture, giant paella pans
>
> Turkey/celebration: minarets in every background, Ottoman-style architecture, oversized Turkish flags
>
> Turkey/eating: Ottoman-style architecture, oversized Turkish flags, exaggerated Turkish carpets
>
> USA/celebration: Oversized American flags, suburban houses with white picket fences, cowboy hats
>
> USA/eating: fast food items, suburban houses with white picket fences, pickup trucks
>
>
> **VLM metrics that inherit similar cultural biases, however these works also use visual perceptual similarity and diversity based metrics that do not use VLMs.**
>
> Our work focuses on cultural relevance, not visual diversity, so diversity metrics (L138-L152) are not applicable to the type of correctness we evaluate. For cultural faithfulness specifically, existing works rely on human annotation, while prior approaches such as [R1] and [R3] rely primarily on VLM-based image–text alignment (e.g., CLIP/SigLIP similarity with “{activity} in {country}”), which we show has near-zero agreement with human judgments for culturally specific activities (Table 2) and therefore inherits the same cultural biases these models encode. Perceptual similarity metrics (e.g., LPIPS[3]) do not address this issue, as they capture low-level appearance diversity rather than culturally grounded correctness, and their encoders are trained on similarly Western-skewed web data [2,3]. While [R3] (concurrent work; ICCV Nov'25) discusses scorer bias (“generative entanglement”), our descriptor-based approach avoids this by not asking a VLM to judge cultural relevance directly; instead, we extract concrete visual elements and compare them to external cultural descriptors annotations, which yields substantially higher agreement with human judgement on faithfulness.

---

> ### Author Response · Authors · 2025-11-26
> **response (part 3/3)**
>
> **Since these MLLMs are also trained on internet data, will they not also inherit similar biases as VLMs?**
>
> Yes; MLLMs show similar bias but less as they utilize an LLM that is trained on text that is more available. While we use MLLMs (InternVL3) to extract object descriptors from images or LLMs like GPT4/Gemini with best cultural understanding [4], we **do not use them directly to evaluate the image**. Instead we use MLLMs to perform generic scene understanding by extracting visible descriptors from the image. Cultural evaluation happens by comparing extracted descriptors to external reference descriptors. In Table 2 we compare our approach with two MLLMs (InternVL3 and QwenVL2.5) and corresponding MLLM-as-a-Judge with the same backbone. Results show AHEaD with InternVL3 backbone achieves
> 0.47 rank correlation (much higher than the 0.20 for InternVL3-as-judge), and AHEaD with QwenVL2.5 achieves 0.42 (vs 0.10 for QwenVL2.5-as-judge baseline on GT-FAITH (overall). This shows that our metric doesn't rely on cultural understanding of the MLLMs
>
> **"exaggeration candidates" (L245), how do we know that these are culturally accurate?**
>
> Related works [1] demonstrated LLMs can effectively identify bias/exaggeration candidates. Building on this, we used LLMs to suggest Exaggeration candidates. We originally validated this in our setting by comparing three approaches for EXAG in Table 11 (Appendix): (1) LLM reference (GT) descriptors, (2) MLLM-extracted descriptors from real images, and (3) exaggeration/Stereotype candidates. Results show exaggeration candidates achieve significantly correlate better with human judgement (e.g., 0.35 vs. -0.27 with LLM GT descriptors on FLUX). This validates that exaggeration candidates capture stereotypical elements, while generic descriptors (whether from LLMs or real images) do not specifically target over-representation patterns.
>
> **When comparing against image-text alignment baselines (Tab 2), what prompts are used, what configuration of CuRe I-T alignment scorer [3], what prompts for MLLMs**
>
> Thank you for constructive feedback, we updated and included the following in A.4; color-coded in blue).
>
> ITA metrics: We use "A photorealistic image of {activity} in {country}" (same as generation prompt).
> CuRe: Since we measure cultural faithfulness (not object recognition), we use their ITA configuration computing average SigLIP2 similarity between images and P(n) = "An image of {activity}" and P\(r) = "An image from {country}". We do not use P\(c) (parent category) as category information already exists in our activity prompts and doesn't apply to social activities.
> MLLM-as-judge: We use the same question asked to human annotators (see Appendix for full prompts).
>
> **Minor Weaknesses**
>
> Thanks, we addressed in the paper (see blue text) and reflect new results:
>
> L117 : Our AHEaD metrics achieve 27\% higher rank correlation with human judgments of cultural faithfulness **compared to using the same MLLM backbone** directly as a judge, and significantly outperform existing image-text alignment metrics.
>
> L118: the best rank correlation with **with human faithfulness** is achieved
>
> **L212: "...maximizing recall of cultural elements" - recall with respect to what? There is no ground truth descriptor set to which the LLM predictions can be compared to?**
> Thank you. We combine descriptors from different LLMs to ensure diverse coverage. We rename "recall" to "coverage" to avoid confusion. We estimate coverage through human evaluation (L349-L355) indicating that our descriptors do not miss major scenarios of possible scenes for activities.
>
> **Do authors instruct annotators on how to "correctly" score or write explanations?**
>
> We do not instruct annotators on what the "correct" score is, as this would bias the study. We only ask for clarification when responses appear contradictory. For example, if an annotator marked most descriptors as incorrect in the relevance question but rated overall descriptor coverage as 4/5 (mostly correct), we asked them to clarify the discrepancy. Note that Prolific (unlike Amazon MTurk) limits rejections to 8 per study, so platform guidelines recommend clarifying inconsistencies through discussion before rejection.
>
>
> **Fig. 3a is hard to visualize; the legend colors do not match, and one metric is better if higher and the other if lower. Consider using a side-by-side bar chart for this.**
>
> Thanks, will update the bar chart for more clarity. We didn't use barchart as covering 16 countries across 2 metrics may take up space. We note "color mismatch" of overlapped is to show both "red"/"blue" simultaneously.
>
>
> [1] OpenBias: Open-set Bias Detection in Text-to-Image Generative Models, CVPR 2024
> [2] No Classification without Representation: Assessing Geodiversity Issues in Open Data Sets for the Developing World
> [3] Does Object Recognition Work for Everyone?
> [4] Culturalbench: A robust, diverse, and challenging cultural benchmark by human-ai culturalteaming

---

### Official Review · Reviewer_RoXn · 2025-11-01

**Soundness:** 2
**Presentation:** 3
**Contribution:** 2
**Rating:** 4
**Confidence:** 4

**Summary:**

The authors introduce CULTIVate, a benchmark of social activities across 16 countries / 9 activity types / 576 prompts / 19k+ generations and propose AHEaD, four descriptor-based, automatic metrics for cultural evaluation: Alignment, Hallucination, Exaggeration, and Diversity. Their composite FAITH score correlates better with human judgments than popular image-text metrics, and they document a consistent Global North > Global South performance gap across SOTA T2I models.

**Strengths:**

1) Interpretable diagnostics: Can list top/bottom descriptors per country/activity to see what is missing or overdone, useful for model iteration
2) No per-image human labels are needed to compute metrics, and it shows good improvements over existing metrics.

**Weaknesses:**

1) Benchmark spans 16 countries, but the human study covers only a smaller subset (7 countries), with no clear rationale or GN/GS/per-country annotation breakdown
2) Descriptors are produced with GPT-4o and Gemini-2.5 Flash, yet “MLLM-as-judge” baselines cover only InternVL/QwenVL - no strong judge like Gemini 2.5 Flash/pro or GPT-4o-vision. Makes it hard to evaluate whether different descriptors are needed and to justify that we need more than one Judge model.

**Questions:**

1) Why not representation from more countries (only 16)? Also, what's the rationale behind surveying only 7 countries?
2) How many annotators rated each image/prompt (per-item replicates)? Also, are the 189 human forms for generated images total across models or per model? If total, the effective sample per model is small, please report per-model/per-item counts.

---

> ### Author Response · Authors · 2025-11-26
> **response**
>
> We thank Reviewer RoXn for their thoughtful review. We are pleased they found our interpretable diagnostics useful, noted that our metrics require no per-image human labels, and highlighted our improvements over existing metrics as well as the consistent Global North vs. Global South gap our benchmark reveals.
> We respond to each specific concern inline.
>
>
> **[W1] … Human study covers only a smaller subset (7 countries), with no clear rationale or GN/GS/per-country annotation**
>
> Thank you for your feedback. GS/GN annotations in the benchmark are based on UN classification [1]. We expanded our human study from 7 to 11 countries (nearly doubling number of prompts; see [G1] and Sec. 4.2). In the original study, 7 countries were representative: 2 GN countries (USA, France) from Americas and Europe, and 5 GS countries (India, China, Iran, Nigeria, Brazil) covering South Asia, East Asia, Middle East, Africa, and Americas. This ensures all major social regions are represented for metric validation.
>
>
> **[W2] … “MLLM-as-judge” baselines cover only InternVL/QwenVL - no strong judge like Gemini 2.5 Flash/pro or GPT-4o-vision**
>
> GPT4o/Gemini are not part of our evaluation method, instead we use InternVL3 as MLLM backbone. Per reviewer suggestion we compared against three MLLMs (GPT4o, InternVL3, and  Qwen2.5-VL): results show our FAITH metric with InternVL significantly outperform InternVL3 and Qwen2.5-VL and achieves comparable performance compared to GPT4o, confirming effectiveness of our descriptor-descriptor framework;   See [G3] from Reviewer rXJH for details.
>
>
> **[Q1] Why not representation from more countries (only 16)?**
> We selected 16 representative countries spanning [2]'s cultural regions, balancing breadth (countries) and depth (activities/prompts per country). Our benchmark is comparable to related work: [3] has 8 countries, [4] has 27 countries across only 4 artifacts, [5] has 64 countries but only 300 prompts total. We plan to expand in future work, though scaling requires extensive computation.
> Please refer to [W1] and [G1] for Reviewer rXJH for details on human study,
>
>
> **[Q2] How many annotators rated each image/prompt (per-item replicates)? ... please report per-model/per-item counts.**
> Thank you for this question. We provide details in L317 and [G1] for Reviewer rXJHE. Specifically:
> Original study: 2 annotators per form, 7 countries, 9 activities per country, 3 T2I models. Hence, 189 unique forms (63 per model) = 378 annotations.
> Expanded study: 2 annotators per form, 11 countries (added 4), 9 activities per country, 3 T2I models. Hence, 381 unique forms (127 per model) = 762 annotations. We updated Tables 2-9 with new data.
> Reliability: Inter-rater agreement (Appendix A.4) is comparable to or exceeds related work's maximum per-country agreement. We observe statistically significant correlations (p << 0.05), confirming our findings are reliable and robust.
>
> [1] https://unctadstat.unctad.org/EN/Classifications/DimCountries_All_Hierarchy.pdf
>
> [2] https://culturalatlas.sbs.com.au/
>
> [3] Beyond Aesthetics: Cultural Competence in Text-to-Image Models
>
> [4] Inspecting the geographical representativeness of images from text-to-image models.
>
> [5] CuRe: Cultural Gaps in the Long Tail of Text-to-Image Systems

---

### Official Review · Reviewer_rXJH · 2025-11-07

**Soundness:** 2
**Presentation:** 1
**Contribution:** 2
**Rating:** 2
**Confidence:** 5

**Summary:**

The authors introduce metrics to effectively evaluate images generated by text-to-image models. The authors first propose building a dataset of 576 prompts from 16 countries, totalling 19000 images. The authors also propose 4 metrics to measure alignment, hallucination, exaggeration, and diversity. The authors conduct a small human study to rate images and compare their metrics with human correlation. They further calculate scores for generated images using their metrics to demonstrate that the disparity between countries in the global north and south, where models perform better on the former.

**Strengths:**

1. The authors introduce new metrics to measure how well T2I models depict cultural activities. As T2I models are being deployed in many parts of the world, this is a very important problem that needs to be tackled, and reliable metrics are essential.
2. The idea behind the metrics is well motivated. I agree that existing works mostly quantify alignment and quality metrics and might miss some nuances. Going beyond these and explicitly calculating hallucination and exaggeration is a nice direction.

**Weaknesses:**

1. **CULTIVate dataset:** The authors introduce a large scale dataset comprising 576 prompts and 19k images, but it is not clear what the utility of the dataset is beyond reporting the AHEaD metrics on them. Many of these image-prompt pairs have no human annotations to compare the correlation of metrics with. Could the authors provide more insights on this?
2. **Human Study:** The authors have conducted a very small human study with a limited number of prompts, i.e, only 9 activities per country for 7 countries, which totals 63 prompts out of 576. Moreover, out of the 6 models used, the authors have conducted the study only for 3 public models. The authors use this to report the correlation between human scores and their metric scores. Due to the small size of the human study, which does not even cover all the countries in the CULTIVate dataset, it is not clear how significant the correlations are. This makes the claim of AHEaD metrics' efficacy weak and the overall study less reliable. I understand that getting human judgments is hard for all the prompts and images, but a representative set of prompts covering all the countries should be used. It would be great if the authors could provide insights on this.
3. **Metric Definitions:**
	1. In L85, the authors contend that existing metrics have a problem in that they use VLM internal knowledge, which inherits biases. But it looks like the author's entire pipeline is based on VLM knowledge. This involves generating descriptors for a prompt and generated images, which are based on the VLM's internal knowledge of the activity. The authors also generate potential stereotype candidates for a prompt which again relies on VLM's knowledge of potential stereotypes that might be biased. This feels a bit contradictory to the initial claim. The authors should either revise the claim or provide more elaboration on why and how this method does not inherit the cultural biases of VLMs.
	2. The core of the metric is to come up with descriptors for a prompt. I am wondering if these capture the full scale of cultural nuances. This is because cultural nuances are also about the values and norms associated with the entities in the images. This can be something like an appropriate gesture, the way people are interacting, or about norms (which people must follow). The descriptors ask for the existence of certain elements, but do they also ask for non-existence? In certain activities, not observing some norms is as important as certain aspects being present. Does the metric account for this? Moreover, a given festival/activity can be celebrated in different ways and have different subcomponents. Generating an exhaustive set of descriptors and comparing with them might not be ideal, as there might be some really valid images that do not cover all possible descriptors generated by the model. Since the metrics divide by the sum of all descriptors, this might underreport the numbers. Hence, I feel this approach does not capture a significant part of cultural nuances. Do the authors have any thoughts on this?
	3. The authors generate stereotype candidates for exaggeration using VLMs and also descriptors. Have the authors verified the accuracy of the different types of descriptors? In L669, the authors mention 90% agreement with humans with their framework. It is not very clear which exact descriptors these agreements are, i.e., whether for alignment, exaggeration. Could the authors provide more details of the human study? Table 9 provides a breakdown, but these are descriptors validated by humans, i.e., precision. Is there any study for recall? It could be possible that there are some essential ones that the LLM missed?
	4. The authors define diversity and semantic diversity, but only report numbers using this. Since the authors introduce this as a new metric, they need to quantify how good this metric is compared to those like the Vendi score and that proposed by [1] and other relevant baselines, which has not been done.
4. **Results:**
	1. The authors have compared their metrics with a limited set of baselines. Some notable baselines that the authors don't evaluate include VIEScore [2], UnifiedReward [3], which have been shown to have good correlation with human judgements [4]. Moreover, the authors have not computed the scores for GPT-4o or Gemini 2.5 (without descriptors, Table 3) to provide scores for faithfulness in Table 2. Only InternVL and QwenVL are used. It is unclear why the authors compare with these models.
		1. A minor point is why QwenVL and InternVL are used for baselines. There is Qwen2.5VL and Intern2.5VL also. Why not use these more powerful models? If the authors have used these, then due to a lack of citation, it is not clear.
		2. The authors should report the numbers of GPT and Gemini with the same prompt of faithfulness that they ask humans for a fair comparison, since they use these models to generate descriptors. [4] already show that these are very good for similar kinds of prompts, and hence it would be an interesting comparison.
	2. It seems from Tables 5,6,7 that the proposed metrics don't correlate very well with humans as compared to baselines like InternVL and QwenVL. It feels like for 2/3 models, QwenVL does better on hallucination (Table 6). Also, for exaggeration, QwenVL does better overall (Table 7). Moreover, it looks like the authors' metrics underperform for the global south, which again could be the result of biases in the descriptors or because of a small human study. This is especially important as authors use this to make claims for disparity between the Global South and North. Could the authors provide more insights into this behavior as this is confusing.
	3. The claim of disparity between Global South and North needs more substantiation. The AHEaD numbers reported are quite close in several cases and hence it would be good to report the mean and standard deviation to quantify how significant these differences are. Also what are the scores provided by humans and how do they differ across the global north and the south?
	4. Why is human-human correlation almost zero for humans for the Qwen model in Table 8? Similarly, in Table 9.
5. **Minor Issues:** Here are some minor writing related errors: L425 (should be do not ask human annotators?), L240 which has an incomplete sentence, Table 7 has wrong numbers bolded. Lack of citations for models in Table 2.

**References:**

[1] Kannen et al. Beyond Aesthetics: Cultural Competence in Text-to-Image Models.

[2] Ku et al. VIEScore: Towards Explainable Metrics for Conditional Image Synthesis Evaluation

[3] Wang et al. Unified Reward Model for Multimodal Understanding and Generation

[4] Nayak et al. CULTURALFRAMES: Assessing Cultural Expectation Alignment in Text-to-Image Models and Evaluation Metrics

**Questions:**

Please see Weakness. I have asked the questions there.

---

> ### Author Response · Authors · 2025-11-26
> **response (part 1/4)**
>
> We thank Reviewer rXJH for their thoughtful review. We are pleased they identified the**the problem important**, found our approach**well-motivated**, and noted our introduction of **new metrics** as **valuable** -- particularly the focus on hallucination and exaggeration, which they highlighted as a **“nice direction.”**
> **Summary of changes to the paper**:
> (1) we have expanded our human evaluation study to include 4 additional countries and more prompts per activity (L315)
> (2) We have split Table 2 into two tables: Table 2 and Table for more clear representation (Hence original Tables 3+ are now Tables 4+)
> (3) We have updated Tables 2,3,6,7,8,9 (new version) accordingly to additional human data
> (4) Included GPT4-o MLLM as a Judge, we've included VIEScore per reviewer suggestion
> (5) Added examples of Image editing (Fig. 8 in Appendix)
> (6) Newly added content/revised content are color-coded in blue
>
> We first address common points across multiple comments of rXJH, then respond to each specific concern inline.
>
> **[G1] Human Study size**
>
> We have expanded our human evaluation to 11 countries (vs. original 7 countries) with 2 prompts per activity (except wedding/funeral as they have only one prompt/subactivity), adding 384 annotations (i.e. Total 762), **nearly 2X the original study** (378 annotations); We'll add the remaining countries in the final version. We've updated Table 2-8 (now Table 3-9) according the new expanded human evaluation data,** results  are consistent with our prior results indicating** the effectiveness of proposed cultural faithfulness metrics:  **ITA baselines show almost 0 agreement** with human judgements, and our proposed **FAITH metric outperform corresponding MLLM baselines significantly** (e.g., FAITH with InternVL3/Qwen2.5-VL outperforms InternVL3/Qwen2.5-VL by up to 0.27/0.32 points).
> We clarify that the original study, while not exhaustive, **covered representative countries**: all social regions in [1], 2 GN countries (regions: Americas, Europe) and 5 GS countries (regions: Africa, Middle East, South Asia, East Asia) and all 9 activity categories. Our goal is building towards a metric that can match human judgment. For metric development, **using a representative subset rather than exhaustive annotation is common practice** (e.g., [7] validated on 35% of their benchmark). **Our inter-rater agreement is comparable to or higher than related cultural benchmarks** that evaluate on full datasets (see A.4), and we observe **statistically significant results** (p<<0.05; e.g., 1.68e-10 on GS for ALIGN). Therefore, while expanding human annotations would be valuable, the **current study size is sufficient and reliable for metric development**.
>
> **[G2] Descriptor Quality/Validation**
>
> As discussed in A.3, we acknowledge reference descriptors are prone to LLM bias. To limit bias, we (1) use best LLMs in cultural understanding according to [4] (2) employed a proposer-refiner method  (see Sec.3), **combining different llms and refining the errors**, **which shows to effectively increase correlation on cultural faithfulness in Table 4. To ensure descriptor completeness, we explicitly generate across five cultural dimensions (Interaction, Setting, Objects, Attire, Spatial), preventing over-focus on a single aspect (e.g., only object oriented descriptors). Our human evaluation shows **90% precision, and high recall** : average of 4.5 out of 5 overall quality score and in 352/378 annotators did not indicate any descriptors were missed. This shows high precision/recall and high quality of llm reference descriptors; details on precision/recall in  Sec 4.2, L352;
>
> We also emphasize that **descriptors not only provide explicit cues, but also  enables interpretable, actionable diagnosis of evaluation scores and cultural failures**: users can use the HAL/EXAG/ALIGN descriptor-based feedback of our frameworks to decide whether to **although effective, prompting LLMs are not the only means to provide descriptors; future works can use human written descriptors or develop RAG/AI agent with search systems to generate descriptors grounded on cultural knowledge bases**

---

> ### Author Response · Authors · 2025-11-26
> **response (part 2/4)**
>
> **[G3] Selection of Baselines**
> Unlike most prior work that relies on human annotations (costly and unscalable) for cultural faithfulness, we aim to build an effective automated metric matching human judgment. Only [2,4] use image-text similarity (CLIPScore/SigLIP) with prompts like "{activity} in {country}", which achieve ** ~0 agreement with human judgment**  (Table 2). We demonstrate that alignment alone is insufficient: adding more cultural elements can increase both alignment and exaggeration simultaneously, decreasing overall faithfulness. Effective metrics ** must negatively correlate with exaggeration and hallucination**  (a property of human faithfulness; Fig. 5), but existing ITA metrics positively correlate with these failures. Hence, our FAITHfulness metric combines ALIGNment, HALlucination, and EXAGgeration, for which we compare against:  (1) ALIGN/FAITH: 5 ITA metrics including the only related cultural metric (CuRe) and 2 MLLM-as-judge baselines. (2) For HAL/EXAG:  ** This is the first work quantifying EXAG/HAL**; we explore both descriptor-based and MLLM-as-judge approaches.
>
> **Comparison with GPT-4o/Gemini MLLM-as-judge:**
>  **We do not use GPT-4o/Gemini in our method, instead we use InternVL3**. This makes InternVL3-as-a Judge a fair baseline. Per reviewer suggestion, we added GPT-4o-as-judge comparisons. Table 2 shows our FAITH with InternVL3/Qwen2.5-VL backbones achieves 0.47/0.42 correlation vs 0.10/0.20 for InternVL3/QwenVL, and is comparable to  GPT-4o (0.48) despite using weaker MLLM. This demonstrates effectiveness stems from the descriptor-based framework, not model capacity.
>
> **[G4] Difference vs CulturalFrames [6]** (concurrent work from EMNLP Nov 2025; it's not fair to be used to undermine our contribution).
>
> Both works independently show existing ITA metrics fail for cultural evaluation. Our work differs significantly: (1) [6] relies on human annotators for cultural faithfulness (costly, unscalable), we propose the first automatic metric for faithfulness (2) We develop methods to measure EXAG/HAL and show their relation with cultural faithfulness   (3) Our benchmarks are complementary: we test more implicit prompts ("greeting in India") compares to [6] ("grandchildren touching feet"). Implicit prompts reveal whether models understand cultural norms independently; explicit prompts test accurate depiction of described scenarios.
>
>
> **utility of dataset beyond reporting AHEaD metrics**
>
> CULTIVate serves multiple research purposes beyond reporting AHEaD metrics: (1) Improving generation quality. We tested using our descriptor-based feedback (HAL/EXAG) to guide image editing. For "Elephant ant man game in Indonesia" (Fig. 1b), HAL identified hallucinated elephants for removal, EXAG identified overrepresented "batik" clothing suggesting more diverse attire. Also, our descriptors + HAL provided reliable context refining images to show correct hand gestures, improving cultural accuracy (examples in Appendix). (2) Developing better metrics. Researchers can correlate their metrics against our EXAG/HAL to measure effectiveness of their metrics, reducing the need for extensive human evaluation (3) Explainable cultural feedback. Users unfamiliar with a target culture can use our descriptor-based feedback to understand specific cultural issues in an image.
>
>
> **Human study … conducted a very small human study …**
>
> Please refer to [G1] for details.
>
>
>  **Use of VLM internal knowledge.**  … existing metrics have a problem in that they use VLM internal knowledge ... The entire pipeline is based on VLM knowledge …
>
> Thank you for this feedback. **The key distinction is that we do not use VLMs to directly score cultural relevance.** Specifically, existing metrics [2,3] compute similarity with prompts like "dragon dance in China," relying on VLMs' internal embeddings on how visually “dragon dance” looks like.  We use MLLMs only for generic scene understanding (asking what do you see in the image). and cultural evaluation happens by comparing extracted descriptors against external reference descriptors, not VLM direct judgment. Results show that  VLM metrics achieve ~0 correlation with human faithfulness while our FAITH metric achieves 0.47 (Table 2).
>
>
> **Descriptor coverage and cultural nuances**. … wondering if these capture the full scale of cultural nuances ... about values and norms …
>
> This is precisely the motivation for our benchmark. We design our benchmark for this goal by focusing on social activities by parsing knowledge bases (e.g. CulturalAtlas and Wikipedia) that include such information (e.g., etiquette, social norms in greeting, traditional games/dances);  We generate descriptors across five dimensions **including interactions/gestures** (e.g., "greeting with bow," "sharing from communal dish") and spatial arrangements (e.g., "seated on floor around sofreh"), capturing behavioral norms beyond visual objects. More details on Descriptor in [G2]

---

> ### Author Response · Authors · 2025-11-26
> **response (part 3/4)**
>
> **Descriptors ask for the existence of certain elements, but do they also ask for non-existence?**
>
> We capture non-existence through the Hallucination (HAL) metric. This approach is more effective than using negative/non-existence descriptors because: (1) negative phrases are unlimited and ambiguous, and (2) models struggle with negation [6]. HAL is effective and increases correlation with Faithfulness (Table 3).
>
>
> **... Generating an exhaustive set of descriptors ... might not be ideal, ... Since the metrics divide by the sum of all descriptors, this might underreport the numbers.**
>
> **Human Evaluation shows high precision/recall, directly addressing this concern** (see [G2] above and/or L350 in paper). Additionally, We address this by generating **mutually exclusive descriptors** that cover diverse valid scenarios for each activity-country pair (E.g., "eating at home in Iran" includes  "eating around table "eating on the floor" which can be covered by our descriptors)
>
> We clarify **ALIGN does not "divide by sum of descriptors." Instead, measures the ratio of covered descriptors** (Eq. 1). We acknowledge coverage over descriptors doesn't become 100% as we intentionally cover diverse possible scenarios of a scene. However, this is not a limitation:
> (1) **Metric is tunable; Users can adjust  for their dataset**  by computing descriptor-descriptor similarity scores similar to Fig.6 in A.3. The threshold (L229) controls forgiveness - lower threshold increases ratio of covered descriptors as a descriptor is decided as covered if its max. similarity score is above the threshold (See Eq.1).
> (2) **Raw scores are not as important as relative ordering; higher alignment indicates higher coverage**. As shown in the table below, our ratio-based metric correlates better with human judgments than alternative designs such as avg similarity (mean over max-similarity to any MLLM descriptor for each LLM descriptor) and top-3 max similarity (average of the three highest per-descriptor similarities). Also, we note that compressed scores is not a unique phenomenon (e.g., [8] indicates CLIPscore operates in [0,0.4] despite being cosine similarity-based)
> We see this work as the first automatic cultural faithfulness metric; rather than optimal descriptor generation or optimal metric. We look forward on future works building on our insights to develop more robust metrics addressing a major limitation in icross-cultural research: over-reliance on human evaluation
> | Method Category        | Backbone   | Spearman (GT-FAITH) |
> |------------------------|------------|---------|
> | MLLM-as-a-judge        | internvl3  | 0.13   |
> | ALIGN (avg similarity)           | InternVL3  | 0.30  |
> | ALIGN (avg top-3 similarity  )     | InternVL3  |0.30   |
> | ALIGN (ratio); Ours    | InternVL3  |0.33
>
>
> **... verified the accuracy of the different types of descriptors? ... Is there any study for recall?**
>
> Human evaluation shows high precision and recall. See [G2] and L352 for details
>
>
> **diversity and semantic diversity ... compared to those like the Vendi score and that proposed by [1] and other relevant baselines**
> We respectfully clarify that **this paper evaluates cultural faithfulness which included (ALIGN,HAL, EXAG, not diversity** (Eq. 5).  Diversity measurements in our framework are secondary analysis tools, fundamentally different from visual diversity metrics** like Vqs  (R1 as cited by reviewer) or perceptual similarity ([3] shows visual diversity doesn't correlate with faithfulness). **Specifically comparing against Vqs is infeasible** as it measures diversity holistically through binary kernels (is country/continent the same?) -- useful for system-level biases, different from our activity-country specific descriptor diversity. In fact, our benchmark shows that images are mostly related to the intended country/continent/artifact which makes Vqs consistently zero regardless of the quality of images. Similarly, two visually different yet culturally unrelated images (i.e. not covering descriptors)  show high visual diversity but low or zero descriptor diversity (no descriptor is covered)
>
> **The authors have compared their metrics with a limited set of baselines …**
>
> See [G3] for detailed baseline discussion. Per reviewer suggestion, we tested **VIEScore with GPT-4o**, **which performed significantly lower than our FAITH metric** (VIEScore: 0.35 overall vs. 0.47  ours rank correlation with GT-FAITH).  We note VIEScore is the top metric in [6], and UnifiedReward underperforms VIEScore and was withdrawn from ICLR'26; hence do not think its critical and will add in future versions due to limited time.

---

> ### Author Response · Authors · 2025-11-26
> **response (part 4/4)**
>
> **Only InternVL and QwenVL are used. It is unclear why the authors compare these models ... report the numbers of GPT and Gemini**
>
> Please refer to [G3]. Briefly: GPT4o/Gemini are **not** used in our evaluate method, we used InternVL3/Qwen2.5-VL as our MLLM backbone. Hence, InternVL3/Qwen2.5-VL are fair baselines. Also, our metric with InternVL3 (overall spearman = 0.47) shows similar performance to GPT4-o (overall spearman = 0.48) (see Table 4/[G3]) but is more interpretable.
>
>
> **Tables 5,6,7 that the proposed metrics don't correlate very well with humans as compared to baselines like InternVL and QwenVL.**
>
> Thank you for the important question; Note Table 5-7 are now Table 6-8.
> We clarify an important distinction: **Tables 5-7 (now 6-8) show individual metric correlations (HAL/EXAG alone), while Table 2 shows our combined FAITH metric (ALIGN+HAL+EXAG). We show  that combining metrics achieves best agreement--individual components naturally show lower correlation.
> **On individual metrics (Tables 6-8)**: We emphasize GT-FAITH as primary ground truth because it directly measures cultural faithfulness. Tables 6-7 show our HAL consistently outperforms MLLM baselines. For EXAG (Table 8), we stress that this is the first work measuring exaggeration and we tested VQAScore and MLLM-as-judge approaches. VQAScore underperforms MLLMs, which in fact supports our main claim that ITA-based metrics are suboptimal for cultural evaluation. We adopt VQAScore in our framework because it enables descriptor-based feedback, though stronger EXAG implementations would improve FAITH precision. We will clarify this in the main text.
>
> **disparity between Global South and North needs more substantiation**
>
> We thank the reviewer for this constructive question. Table 1 shows this disparity is consistent across all T2I models and metrics (higher ALIGN, lower HAL/EXAG on GN). We also observe higher variation (standard deviation) on GS, confirming the GN-GS gap. The slightly smaller gap between GS/GN (vs object-centric benchmarks) is justified: Unlike objects, social activities are compositional -- only certain aspects might be depicted incorrectly (e.g., exaggerated attire), causing smaller GS/GN gaps compared to object-oriented categories where the entire object is typically wrong. Additionally, many activities (e.g., wedding, eating) have cross-cultural overlap and are more generic, and T2I models perform well on both GS/GN. T2I models may fail on culture-specific activities even in GN countries (e.g., misrepresenting "tute," a Spanish card game, as eating dinner). In this work, human evaluation's purpose is to evaluate metric-human agreement, not exhaustive cultural evaluation; hence, only a subset of prompts are selected, making raw scores not representative (see Appendix for human scores). We demonstrate that our metrics reliably capture cultural nuances, through this human agreement, and once we’ve verified this, we provide the more exhaustive cultural evaluation through the automated metrics metrics, as they are more scalable. They can also be used to provide information about a new country for which no human evaluation data is available.
>
> **Why is human-human correlation almost zero for humans for the Qwen model in Table 8?**
> Human-human correlation is 0.40 on GT-FAITH (Table 8; now Table 9) after expansion. We report H-H correlation as reference but use Krippendorff's α (see A.4) for inter-rater agreement as it is more appropriate for Likert scales. Per-country agreement ranges from 0.16-0.62, exceeding [6]'s maximum (0.42). We recomputed H-H Spearman on EXAG (Table 11): 0.26, 0.32, 0.33 for FLUX, Qwen, and Stable Diffusion respectively. We will update Table 11 with additional human data in the next version.
>
> **minor ... There is Qwen2.5VL and Intern2.5VL. Why not use these more powerful models?**
>
> Thanks, we do use two powerful models, namely InternVL3-14B and Qwen2.5-VL-7B-Instruct (L727, L711). We've clarified model versions in the tables (blue color).
>
>
> **Minor: ... writing related error...Table 7 has wrong numbers bolded..Lack of citations for models in Table 2.**
>
> Thanks, we've fixed these issues in the current version.
>
>
> [1] https://culturalatlas.sbs.com.au/
>
> [2] An image speaks a thousand words, but can everyone listen? On image transcreation for cultural relevance (EMNLP 2024)
>
> [3] CuRe: Cultural Gaps in the Long Tail of Text-to-Image Systems (ICCV 2025; Oct 2025)
>
> [4] Culturalbench: A robust, diverse, and challenging cultural benchmark by human-ai culturalteaming
>
> [5] This is not a Dataset: A Large Negation Benchmark to Challenge Large Language Models, EMNLP 2023
>
> [6] CulturalFrames: Assessing Cultural Expectation Alignment in Text-to-Image Models and Evaluation Metrics
>
> [7] Evaluating Text-to-Visual Generation with Image-to-Text Generation
>
> [8] CLIPScore: A Reference-free Evaluation Metric for Image Captioning

---

### Author Response · Authors · 2025-12-03
**Summary  1/3**

Given changes in the review process, we summarize:  **our paper's contributions**, **strengths highlighted by reviewers**, and **changes in the rebuttal and how we addressed the comments**

### **What this paper is.**
- **CULTIVate benchmark.** **we propose to evaluate cultural understanding through social activities** (greeting, games, dance, eating etiquette), which **more rigorously test culture compared to prior object-centric studies**. Activities require depicting interactions, social norms, and contextually appropriate scenes and objects. CULTIVate is grounded in knowledge bases (CulturalAtlas, Wikipedia), includes 576 prompts across 16 countries from all socio-cultural regions in CulturalAtlas. We find consistent Global North/Global South disparity, and reveal a new failure mode: T2I models often depict correct concepts but include hallucinated or exaggerated elements -- unlike object-centric studies where objects are right or wrong.
- **AHEaD diagnostic tool to diagnose cultural faithfulness**: We developed a set of interpretable descriptor-based tools that quantitatively evaluate cultural ALIGNment, HALlucination, EXAGgeration, and measure Descriptor Diversity.  We find that **alignment alone is insufficient**, and an **effective cultural faithfulness metric must negatively correlate with hallucination/exaggeration** as an over-exaggerated image or image with out-of context/incorrect objects is less relevant to the target culture . For instance, an exaggerated image includes too many cultural elements (e.g., too many rugs at restaurants in Iran), this makes the image unrealistic and not faithful to the culture.

- **FAITH metric**. First automatic cultural faithfulness metric combining ALIGN, HAL, and EXAG. This addresses a key gap: existing works rely on costly, unscalable human evaluation. [1,2] use image-text similarity as proxy, but we show these ITA methods exhibit less than 0.15 correlation with human judgment and **do not** negatively correlate with exaggeration/hallucination (i.e., ITA  incorrectly score higher faithfulness when images over-represent cultural elements or include out-of-context objects)

### **Strengths noted by reviewers.**

The problem is **important** and **underexplored** (rXJH, b2M6), and the paper offers a benchmark that meaningfully expands cultural evaluation into social activities rather than object-centric domains (b2M6). The **AHEaD metrics were highlighted as a key contribution**: reviewers found them well **motivated** (rXJH, yiqA), calculating hallucination and exaggeration as a **nice direction** (rXJH), with **interpretable** descriptor-based feedback (b2M6, RoXn, yiqA). They also noted metrics are **actionable** (yiqA), useful for model iteration (RoXn), and **well defined** (yiqA). Reviewers noted that **no per-image human labels are required** (RoXn). Multiple reviewers found that FAITH **correlates better with human judgments** than image-text similarity or MLLM judges (b2M6, yiqA, RoXn). The writing was described as **clear** and **well presented** (b2M6, yiqA), and the **ablations and analyses** were noted as **comprehensive** and **informative** (yiqA).

---

> ### Author Response · Authors · 2025-12-03
> **Summary 2/3**
>
> **Human study scale and validity** (rXJH, RoXn, yiqA).
> - We expanded the human study from 7 to 11 countries with 2 prompts per activity, nearly doubling annotations (378 to 762).
> - Human Study spans all socio-cultural regions from CulturalAtlas : **Global North (4)**: USA, Spain, Germany, France; **Global South (7)**: Iran, Turkey, China, Nigeria, Brazil, Mexico, India
> - Per-country Krippendorff's Alpha and Cohen's k **match or exceed related cultural benchmarks' highest agreement** and correlations are statistically significant (p << 0.05): **confirming study reliability** and **correctness of findings**.
> - **We've updated Tables 2,3,6,7,8,9 to include additional human evaluation**, and observe that **Results/findings of expanded evaluation remain consistent with original human evaluation**: (1) Image-text alignment metrics show ~0 correlation with human judgments, demonstrating ineffectiveness for cultural faithfulness. (Table 2) **(2) FAITH with InternVL3/Qwen2.4-VL significantly outperforms MLLM-as-judge baselines (+0.27 InternVL3, +0.32 Qwen2.5-VL)**, showing descriptor-based grounding is essential. (3) **Best correlation happens when combining ALIGN,EXAG, HAL (Table 3)**
>
> **Reference Descriptor validity, bias** (rXJH, b2M6) and **their completeness** (rXJH).
> - As already acknowledged in A.3, LLM-generated reference descriptors are prone to bias. We clarified that we took multiple steps to mitigate this: (1) Developed proposer-refiner pipeline combining multiple LLMs to increase diversity and refined final output to increase precision; Table 4 shows this is effective.  (2)  Used best-performing LLMs for cultural understanding according to [3]
> - To **ensure completeness**we explicitly generated descriptors to be **mutually exclusive** (covering different possible scenarios of each activity) across 5 dimensions (attire, background, objects, interaction/gesture, spatial)
> -  Human validation confirms effectiveness: **>90% precision** and **High recall:  4.5/5 coverage score (352/378 annotations found no missing descriptors/scenario) **
> **Fair MLLM-as-a-Judge baseline** (rXJH,RoXn)
> - We clarified: **InternVL3-as-judge is the fair baseline**.  GPT-4o/Gemini are **not** part of our evaluation method --- we use InternVL3 to extract image descriptors. GPT-4o/Gemini only generate reference descriptors offline during benchmark creation  (no images).
> - Per request, Added GPT-4o-as-judge (Tab. 2): 0.48 correlation with human judgments. **FAITH with InternVL3 matches this (0.47), proving framework effectiveness over model capacity.**
> **Bias of MLLMs** (b2M6)  and **results across different extractors** (yiqA)
> -  Unlike MLLM-as-judge (direct evaluation), we use MLLMs only for generic scene understanding-- extracting visible elements. Cultural evaluation compares these against external reference descriptors, avoiding reliance on MLLM cultural biases.
> - Tested with two independent extractors (InternVL3, Qwen2.5-VL)**Both show strong, stable performance (0.47/0.42) and significantly outperform their corresponding MLLM baseline (0.20/0.10)** (Table 2). Similar trend is observed on HAL (Tab . 6) which shows robustness of our method as mentioned by yiqA
>
> **GN/GS disparity significance** (rXJH);
> - Added standard deviation to Table 1.Results show **consistently higher variation on GS and consistent gap between GS and GN across all metrics/models**, confirming systematic disparity.

---

> > ### Author Response · Authors · 2025-12-03
> > **Summary 3/3**
> >
> > **Novelty of Benchmark & New Failures (exaggeration/hallucination)** (b2M6)
> > - We strongly disagree about the relevance of papers cited. We clarified: While some benchmarks include **minimal**  activity prompts, their focus is primarily object-centric (e.g., [5] evaluates 3 "cultural objects": landmarks, architecture, apparel).  We provided per-paper comparison in rebuttal and will clarify distinctions in Section 2 (most papers are already cited). We note that **b2M6** has already acknowledged this distinction of our benchmark in Strengths: "new benchmark that is focused on cultural activities in contrast to prior work that focuses more on cultural objects"
> > - We clarified: **We are first to quantify exaggeration and hallucination**, and propose a cultural faithfulness metric
> > - **While stereotypes may cause exaggerated images, they are distinct phenomena**. Stereotypes measure **systematic bias across distributions** (e.g. 80% of Iranian men have beared, certain nationality is not happy); exaggeration measures **over-representation within individual images** (e.g., too many rugs, etc. ).  A distribution can be stereotypical with realistic individual images, or a single image can be exaggerated without reflecting a broader stereotype.
> > **Exaggeration candidates effectiveness** (b2M) and **details on how to generate them ** (yiqA):
> > -  Table 11 (already included in original submission) **validates effectiveness of exaggerated candidates** by comparing against alternatives designs. Exaggeration candidates achieve 0.35 correlation vs. -0.27 for reference descriptors (FLUX), confirming they capture over-representation.
> > - Clarified generation method and added the prompt in Appendix (Table 27).
> > **Per country & per activity results** (yiqA)
> > - Pointed to existing figures: 3a (per-country), 3b (per-activity), 4 (per-dimension).
> > **Inter-rater agreement and p-value** (yiqA)
> > - Measured per-country Krippendors Alpha and Cohen k and included results in A.3: **we observe inter-rater agreement match or exceed prior works**. We also computed p-value for our metric in Table 1 and2, and observed all p-values <<0.05 (e.g. 1.68e-10 for ALIGN in GS)
> > **Limited baselines in Table 1** (rXJH)
> > - We respectfully disagree that our baselines in Table 1 are limited.**We already had 7 baselines**: 5 ITA metrics, including Cure [1]  (only recent cultural metric appeared in ICCV'25 in October) and VQAscore (sota metric) and 2 MLLM-as-a Judge. VIEScore[6] was only used in [4], concurrent work appeared in EMNLP'25 in November.
> > - Per reviewer request, we included VIEScore with GPT4o (table 1) and results show that significantly underperforms our metric with InternVL3 and GPT4o baseline itself, confirming ineffectiveness of these metrics.
> >
> > Our rebuttal addresses all reviewer concerns through expanded evaluation, additional baselines, and clarifications. This work fills an important gap by providing the first automatic cultural faithfulness metric with interpretable diagnostics. The paper can be updated with rebuttal content and minor revisions and **no new experiments are needed**.
> >
> > [1] CuRe: Cultural Gaps in the Long Tail of Text-to-Image Systems, ICCV, October 2025
> > [2] An image speaks a thousand words, but can everyone listen? On image transcreation for cultural relevance
> > [3] Culturalbench: A robust, diverse, and challenging cultural benchmark by human-ai culturalteaming
> > [4] CulturalFrames: Assessing Cultural Expectation Alignment in Text-to-Image Models and Evaluation Metrics

---

### Meta-Review · Area_Chair_bzmY · 2026-01-06

**Summary:**

The paper introduces a benchmark focused on evaluating the cultural faithfulness of text-to-image (T2I) models through the lens of social activities (e.g., greetings, dining, games) rather than traditional object-centric categories. The authors propose AHEAD, a suite of four automated, descriptor-based metrics to measure Alignment (ALIGN), Hallucination (HAL), Exaggeration (EXAG) and Diversity.
A central contribution is the FAITH metric, which combines alignment, hallucination and exaggeration to provide an automated measure that correlates more strongly with human judgment than existing image-text alignment (ITA) metrics. The study reveals systematic performance disparities and shows that models generally perform better for Global North countries than for the Global South.

The authors provided a thorough rebuttal that substantially strengthened the paper. They doubled the human study size (from 378 to 762 annotations) and expanded it to 11 countries, confirming that their results remained consistent. They successfully demonstrated that their descriptor-based framework (FAITH) significantly outperforms direct MLLM-as-judge baselines, showing that the effectiveness comes from the methodology rather than just the model capacity.

**Reviewer Concerns:**

The reviewers initially raised several significant concerns that the authors have largely addressed through extensive rebuttals and paper revisions.

**Reviewer Scores:**

While Reviewer rXJH (initially with 2 score) remained highly critical, the authors pointed out factual inaccuracies in the reviewer's characterization of concurrent work. I believe the characterizations of different studies are quite subjective and should not hurt the novelty of this study.

Reviewers were initially divided, but the sentiment could get shifted towards accept with the expanded evaluations during the rebuttal period.

---

### Decision · Program_Chairs · 2026-01-26

Accept (Poster)